# Non-Gaussian data assimilation of satellite-based Leaf Area Index observations with an individual-based dynamic global vegetation model

Hazuki Arakida[1], Takemasa Miyoshi[1, 2, 3], Takeshi Ise[4], Shin-ichiro Shima[1, 5], and Shunji Kotsuki[1]

[1]RIKEN Advanced Institute for Computational Science, Kobe, 650-0047, Japan
[2]Department of Atmospheric and Oceanic Science, University of Maryland, College Park, MD 20742, USA
[3]Application Laboratory, Japan Agency for Marine-Earth Science and Technology, Yokohama, 236-0001, Japan
[4]Field Science Education and Research Center, Kyoto University, Kyoto, 606-8502, Japan
[5]Graduate School of Simulation Studies, University of Hyogo, Kobe, 650-0047, Japan

*Correspondence to*: Hazuki Arakida (hazuki.arakida@riken.jp) and Takemasa Miyoshi (takemasa.miyoshi@riken.jp)

**Abstract.** We developed a data assimilation system based on a particle filter approach with the Spatially Explicit Individual-Based Dynamic Global Vegetation Model (SEIB-DGVM). We first performed an idealized observing system simulation experiment to evaluate the impact of assimilating the leaf area index (LAI) data every 4 days, simulating the satellite-based LAI. Although we assimilated only LAI as a whole, the tree and grass LAIs were estimated separately with high accuracy. Uncertain model parameters and other state variables were also estimated accurately. Therefore, we extended the experiment to the real world using the real Moderate Resolution Imaging Spectroradiometer (MODIS) LAI data, and obtained promising results.

## 1 Introduction

The terrestrial biosphere is an important part of the Earth System Model (ESM) to simulate the carbon and water cycles. However, terrestrial biosphere models tend to have large uncertainties, for example, in phenology (Richardson et al., 2012; Murray-Tortarolo et al., 2013) and in spatial distributions of plant species (Cheaib et al., 2012). Recently, data assimilation (DA) methods which incorporate observation data into models have been applied to terrestrial biosphere models to reduce the uncertainties in the state variables and model parameters (Luo et al., 2011; Peng et al., 2011). Previous studies have successfully applied the ensemble Kalman filter (e.g., Evensen, 2003; Williams et al., 2005; Quaife et al., 2008; Stöckli et al., 2011) or adjoint method (e.g., Kaminski et al., 2013; Kato et al., 2013) to the *static* vegetation models, but studies with the *dynamic* global vegetation models (DGVMs) are still limited (Luo et al., 2011; Peng et al., 2011), although Hartig et al. (2012) pointed out the importance.

The *static* vegetation models are time-independent and do not include the vegetation succession process (Peng, 2000). Alternatively, DGVMs include the vegetation succession process and can simulate carbon and water cycle changes linking to the vegetation shift under the changing climate. Especially, *individual-based* DGVMs simulate local interactions among individual plants such as competitions for light and water, so that the model can simulate the vegetation succession more explicitly (Smith et al., 2001; Sato et al. 2007). Garetta et al. (2010) pioneered to apply DA to an *individual-based* DGVM for

paleoclimate, but no study has been published thus far to assimilate fine time-scale data from satellites and ground stations using an *individual-based* DGVM. If the initial vegetation structure and the model parameters of an *individual-based* DGVM are estimated more accurately by assimilating the fine time-scale data, the uncertainties of the simulated future vegetation would be greatly reduced.

This study explores to assimilate frequent satellite-based Leaf Area Index (LAI) data with an *individual-based* DGVM known as the SEIB-DGVM, standing for Spatially Explicit Individual-Based DGVM (Sato et al., 2007). We developed a non-Gaussian ensemble DA system with the SEIB-DGVM based on a particle filter approach. Although the particle filter is an existing, well-known approach, this is the first attempt to apply it to an *individual-based* DGVM with frequent LAI data. Therefore, we focus on the methodological development in this study and perform a series of numerical experiments at a single

location with only a couple of plant functional types (PFTs) as the first step. It would be numerically straightforward to extend it to the global scale in the future studies, since the local-scale experiments can be performed in parallel for different locations. In the present study, we first perform idealized simulation experiments to investigate how well we can estimate the model parameters associated with phenology by assimilating the LAI data every 4 days, simulating the satellite-based LAI product from the Moderate Resolution Imaging Spectroradiometer (MODIS) aboard the Terra and Aqua spacecraft. We also investigate

to what extent assimilating the LAI data could improve the estimates of the state variables such as GPP (Gross Primary Production), RE (Ecosystem Respiration), NEE (Net Ecosystem Exchange), and biomass, the most fundamental variables for carbon cycle and vegetation states. Sensitivities to the filter settings such as the random perturbation sizes and particle sizes are also investigated. Following the idealized experiments, we perform an experiment using the real MODIS LAI observation data to see how well the proposed approach performs in the real world.

**2 ~~Methods~~Method**

**2.1 SEIB-DGVM**

    The SEIB-DGVM simulates establishment, growth, and decay of the individuals of prescribed PFTs within a spatially explicit virtual forest (Sato et al., 2007), forced by climate conditions such as air temperature, soil temperature, cloudiness, precipitation, humidity, and winds. We used version 2.71 (Sato and Ise, 2012) but with minimal modifications for DA. The

model simulates daily states, but the original model outputs were only once per year. Outputs are needed for DA every once in 4 days, so that we modified the model code to output the model states every 4 days. In addition, the original model code assumed running for many years continuously, and the initial seed for the random number generator was fixed. Since in this study we stop the model every 4 days, and the same seed is repeated every time when we start the model. Therefore, we modified the model code to randomly generate the seed for the random number generator every time when we initiate the

model. Other modifications are summarized in Appendix.

    The size of the model state space is determined by the prognostic variables for tree, grass, forest as a whole, and soil. Each individual tree has 13 prognostic variables such as biomass of root, leaf and trunk, and we assume that up to 300 trees can exist

in the forest area. Therefore, the number of tree variables is less than or equal to 3900 (i.e., 300 x 13). As for grass, the forest area is divided into 30 by 30 grid cells, and each grid cell has 4 variables such as biomass of root and leaf. Hence, the number of grass variables is fixed at 3600 (i.e., 30 x 30 x 4). In addition, forest as a whole has 8 prognostic variables such as snow and soil carbon mass, and finally, soil moisture (1 variable) is defined for 30 soil layers. Therefore, the number of state variables is between 3638 (no tree, i.e., 0 + 3600 + 8 + 30) and 7538 (300 trees, i.e., 3900 + 3600 + 8 + 30).

Among the various model outputs ranging from individual tree height to soil water content (Sato et al., 2007, with updated information available from the package of version 2.71), we focus on LAI because it is a key to the vegetation model, and because previous studies show a promise in assimilating satellite-based LAI data with a *static* vegetation model (Stöckli et al., 2011) and a *non-individual-based* DGVM (Demarty et al., 2007). We extend the previous studies to assimilate the LAI data with the *individual-based* DGVM.

## 2.2 Particle filter-based DA

*Individual-based* DGVMs include highly nonlinear processes such as occasional establishment and death of individual plants. These processes produce and eliminate state variables, and the phase space changes time to time. DA methods that have been used in geophysical applications usually assume that the state variables are defined uniquely for the given dynamical system and that the phase space dimension stays the same. The widely-used ensemble Kalman filter, for example, finds the best linear combination of the ensemble with optimal fit to the observations, but it is not trivial to define a linear combination or even the ensemble mean for the variables missing in some ensemble members. Therefore, it would not be trivial to apply the widely-used DA methods to *individual-based* DGVMs.

Alternatively, particle filters run independent parallel simulations or particles and represent the probability density function (PDF) explicitly by assigning probability to each particle. Therefore, particle filters can handle non-Gaussianity and nonlinearity explicitly, and can be applied to the *individual-based* DGVMs in a straightforward manner (e.g., Garetta et al., 2010) even though the phase space dimension is different for each particle.

Here we adopt a particle filter approach known as the Sequential Importance Resampling (SIR, Fig. 1) (Gordon et al., 1993). Although the method is not efficient for large dimensional systems, (e.g., Bickel et al., 2008; Snyder et al., 2008; 2012; 2015), we tested this well-known method as the first attempt to construct the DA system with SEIB-DGVM. First, $n$ parallel simulations are performed, and each simulation is considered as a particle representing the true state of the system with equal probability. Next, likelihood $l_t^{(i)}$ is calculated for each particle using the Gaussian likelihood function:

$$l_t^{(i)} = p(y_t | x_{t|t-1}^{(i)}) = \frac{1}{\sqrt{2\pi \cdot \sigma^2}} \exp\left\{ -\frac{(y_t - x_{t|t-1}^{(i)})^2}{2 \cdot \sigma^2} \right\} \quad for\ i = 1, \dots, n. \quad (1)$$

Here, $x_{t|t-1}^{(i)}$ denotes the simulated LAI of the $i$th particle at time $t$ from the previous time step $t$-1, $y_t$ the observed LAI at time $t$, and $\sigma$ the observation error standard deviation. Since the prior probability is uniform, Bayes' rule gives that the posterior probability of the $i$th particle is proportional to $l_t^{(i)}$, i.e., the particles closer to the observation have more probability. Next, we

resample the particles, so that each particle has equal probability. The particles with more probability (larger $l_t^{(i)}$) are duplicated, and the particles with less probability (smaller $l_t^{(i)}$) are removed. If $n$ is sufficiently large, we can evaluate the posterior PDF accurately. Each resampled particle represents the true state of the system with equal probability and acts as the initial particle for the next time step. This Bayesian framework is repeated.

## 2.3 OSSE and the real-world experiment

We first perform a series of idealized Observing System Simulation Experiments (OSSEs). The OSSE (e.g., Atlas, 1997) is a widely-used approach in meteorological DA to test the general performance of a DA system and to evaluate the impact of specific observing systems. OSSE has the nature run, which is usually generated by running a simulation for a certain period. Observation data are simulated from the nature run by applying the observation operator, i.e., converting the model variables to the observed variables. Here, we add artificial random noise to simulate the observation error. DA experiments are initiated from the state independent of the nature run, and the simulated observations are assimilated. The resulting analyses and subsequent forecasts are compared with the nature run to evaluate the performance of DA. Once an OSSE is done, it is straightforward to extend the OSSE to the real world by simply replacing the simulated observations with the real-world observations.

## 3 OSSE

### 3.1 Experimental design

To generate the nature run, the SEIB-DGVM was initialized with the bare ground (i.e., no plant at the beginning) and was run for 107 years using the climate forcing data from year 2001 to 2010 available at the SEIB-DGVM webpage (http://seib-dgvm.com/). Here, the 10-year forcing data are repeated for the 107-year simulation, and the last 7 years from year 101 to 107 use the actual climate forcing of 2001 to 2007, so that we call year 101 to 107 to be 2001 to 2007. The daily climate data were generated by the procedure of Sato and Ise (2012) with updated information available at the SEIB-DGVM webpage, based on the monthly Climate Research Unit observation-based data (CRU-TS3.22 0.5-degree monthly climate time series) (Harris et al., 2014) and the daily data from the National Centers for Environmental Prediction (NCEP)/National Center for Atmospheric Research (NCAR) reanalysis (Kalnay et al., 1996). We chose the study area at one of the AsiaFlux sites, the Siberia Yakutsk Larch forest site at Spasskaya Pad, the middle basin of River Lena (62º 15' 18" N, 129º 14' 29" E). The observed climate data at this site were not directly used in this study, but these data may have been included in the NCEP/NCAR reanalysis. Field observed carbon flux data are available as the ground truth to verify the DA results at this site. Forced by the climate data, the SEIB-DGVM simulates the vegetation shifts from the bare ground to a grassland, and then to a forest. The two PFTs, the boreal deciduous needle leaved trees and C3 grass, are the dominant PFTs in this study area. Therefore, we do not consider the other PFTs in this study following Sato et al. (2010). We call these two PFTs simply "tree" and "grass".

The nature run (Fig. 2 a) was performed with the "true" parameter values Pmax = 15 $\mu molCO_2$ m$^{-2}$ s$^{-1}$ and Dor = 230 DOY (day of year) for tree and Pmax = 9 $\mu molCO_2$ m$^{-2}$ s$^{-1}$ and Dor = 270 DOY for grass, where Pmax and Dor stand for the maximum photosynthesis rate and the start date of the dormancy, respectively (Fig. 2 b). Hereafter we omit the units for Pmax ($\mu molCO_2$ m$^{-2}$ s$^{-1}$) and Dor (DOY) for simplicity. The LAI observations for the last 4 years from year 2004 to 2007 were created by adding independent Gaussian random noise to the LAI values from the nature run (Fig. 2 a) every 4 days, simulating the MODIS LAI product. Here, the observation error standard deviation was given by 10 % of the nature run LAI value. The observed LAI < 0.5 were not used for DA because the MODIS data for the real-world experiment did not include LAI < 0.5. There are too few data with real MODIS LAI < 0.5, and we assign the missing value in preprocessing. Since the LAI is observed only when 0.5 or larger, the LAI observation exists only in the summer season.

Next, 8000 particles (parallel simulations) were generated with uniformly perturbed parameters: Pmax = [0, 60] for tree, Pmax = [0, 15] for grass, Dor = [200, 300] for both. Here, [*a*, *b*] denotes random draws from the uniform distribution between *a* and *b*. These initial perturbation sizes are based on the previous studies (Kolari et., 2006; Zeng et al., 2011; Zhao et al., 2015; Takagi et al., 2015). We ran 8000 parallel simulations for 103 years for spin-up from the bare ground using the same climate forcing data as the nature run. In the course of the vegetation succession, these randomly perturbed parameter sets result in a variety of LAI simulations (Fig. 2 b).

The 8000 particles at the end of the 103-year spin-up runs are used as the initial conditions for DA. The simulated LAI observations are assimilated every 4 days. The nature run and particle filter use the same climate forcing data, so that the difference comes from the model parameter values. The particles continue to be the free runs until the first LAI observation is assimilated in the summer season. The state variables and model parameters are estimated together at DA, and the model systematic errors associated with the model parameters are corrected by DA with parameter estimation. No explicit bias correction is applied. To avoid the exact duplications after resampling, the model parameters Pmax and Dor are randomly perturbed for the duplicated particles. The random perturbations avoid particle degeneracy, which usually causes filter divergence. After some tuning, we found proper perturbation sizes that work for stable filtering without causing particle degeneracy, especially for biomass which is found to be the most sensitive to the perturbation sizes. Here, random draws [-4, 4] are added to Pmax for tree and to Dor for both tree and grass, and [-1, 1] are added to Pmax for grass because the initial Pmax perturbation size for grass is a quarter of that of tree. The sensitivity to the resampling perturbation sizes will be discussed in the next session. In case that these perturbed parameters exceed the corresponding initial parameter range, the excess value was bounced back from the limits. To assess the impact of DA, we also perform an experiment without DA ("NODA" hereafter), and compare to the experiment with DA ("TEST" hereafter).

**3.2 Results**

Figure 3 shows the time series of LAI for NODA (left) and TEST (right). The observations (Fig. 3 a, blue dots with error bars) cannot distinguish the tree and grass, but the model simulates LAIs for tree and grass separately (Fig. 3 b, c). Although the particles without DA are widely spread (left, gray areas), DA makes the particles much narrower (right) and consistent

with the nature run (red curves). With DA, the median of the particles for tree is almost identical to the nature run for the entire 4 years (Fig. 3 b, right). As for grass, the median of the particles is also very close to the nature run with DA, but in the first three years the dormancy period is delayed (Fig. 3 c, right).

The model parameters are estimated accurately (Fig. 4). There is no direct observation of these parameters, so that the estimations are purely due to DA of the LAI observations. Although the particles of the NODA experiment are uniformly distributed (Fig. 4, left), DA makes the particles close to the true parameters (Fig. 4, right). Since we assimilated the LAI only when 0.5 or larger, DA has an impact only in the summer season when the leaves grow. It takes 1-4 years until the true values fall within the quartiles of the particles. The Pmax estimates for both tree and grass show occasional jumps, but tend to stay around the true values (Fig. 4 a, b). Dor for tree seems the most accurate and stable after the dormancy period of the first year (Fig. 4 c). Dor for grass takes the longest; the estimation is not accurate until the dormancy of the fourth year (Fig. 4 d). This may be related to the previous results showing the erroneous estimates of the grass LAI near the dormancy period in the first 3 years (Fig. 3 c). The systematic errors in NODA come from the uncertain parameter settings. TEST can estimate the parameters through DA, and can reduce the systematic errors. This is different from the bias-correction strategy of the first guess.

Other model variables such as GPP, RE, NEE and biomass show large improvements (Fig. 5). Although the particles of the NODA experiment are widely spread, DA with only LAI observations greatly reduces the uncertainties for the four variables, and the estimations are generally reasonable.

## 4 Sensitivity experiments for OSSE

### 4.1 Sensitivity to the nature run

To investigate the sensitivity to the choice of the nature run, we performed two additional OSSEs, which we call "OSSE2" and "OSSE3", by generating different nature runs with different parameter sets (Table 1). The random numbers for the observation errors are also different. The other settings follow the TEST experiment.

The results show that both OSSE2 and OSSE3 perform well in general. Namely, the LAI and parameters are estimated generally well (Fig. 6). We find the main difference between OSSE2 and OSSE3 in the parameters for grass (Fig. 6 c, e). OSSE3 shows significantly larger uncertainties for the parameters for grass. In OSSE2, the Pmax value for grass is larger and produces more grass LAI. Since grass starts to grow earlier and stays longer than tree, it is critical to have LAI observations near the emerging and falling periods for estimating the grass parameters. Due to the larger Pmax value for grass in OSSE2, LAI can be observed with the observing threshold of LAI = 0.5 near the emerging and falling periods. By contrast, in OSSE3, the Pmax value for grass is smaller, and the small grass LAI < 0.5 cannot be observed. We can see this in the LAI time series (Fig. 6 a, right) near the tails in the spring and fall seasons every year. The uncertainties of LAI are not reduced year by year,

corresponding to the large uncertainties of the grass parameters. In the summer, LAI becomes larger mostly due to trees, so that the tree parameters can be estimated well.

## 4.2 Sensitivity to the initial perturbation size

Here we investigate the sensitivity to the initial perturbation sizes with particle sizes ranging from 1000 to 16000. Table 2 shows the three initial perturbation settings: small, moderate and large. For the TEST experiment, the moderate initial perturbation sizes were used. We perform additional sensitivity experiments with the small and large initial perturbation sizes. Except for the initial perturbation sizes and the particle size, the experiments follow the TEST experiment.

Tables 3 show the mean absolute errors (MAE) and the widths of the 1-99% quantiles, respectively, averaged over a year in 2007. We consider that the filter diverges when the MAE is larger than the half width of the 1-99 % quantiles, as shown by gray shades in the tables. The results show that the filter diverges for biomass in 10 out of 15 experiments. The 5 experiments that do not diverge are (4000; small), (8000; small), (16000; small), (8000; moderate) = TEST, and (16000; moderate), where ( ; ) denotes (particle size; initial perturbation sizes). (1000; large) causes filter divergence for most variables and parameters. (2000; large) shows filter divergence for Dor for grass in addition to biomass. Sampling a wider interval with a smaller particle size generally reduces the particle density, or the effective number of the particles, so that the results seem to be reasonable.

## 4.3 Sensitivity to the resampling perturbation size

Here we investigate the sensitivity to the resampling perturbation sizes with particle sizes ranging from 500 to 16000, in a similar way as the previous subsection. Resampling perturbations add random perturbations to Pmax and Dor when resampling and avoid particle degeneracy. Table 4 shows the three resampling perturbation settings: small, moderate, and large. For the TEST experiment, the moderate resampling perturbation sizes were used.

Tables 5 show similar tables as Tables 3 but for the sensitivity to the resampling perturbation sizes. We use the similar notation of ( ; ) denoting (particle size; resampling perturbation setting). The results show that the filter diverges for biomass in 13 out of 18 experiments. The 5 experiments that do not diverge are (4000; large), (8000; moderate) = TEST, (8000; large), (16000; moderate), and (16000; large). (500; small) is most unstable, with more variable and parameter showing filter divergence. Resampling perturbations act as variance inflation in the ensemble filters (e.g., Anderson and Anderson 1999). It is known that variance inflation generally stabilizes the filter, and the results obtained here seem to be consistent. With 4000 particles or more, the parameters and state variables except for biomass were estimated accurately, although the filter collapsed for biomass with smaller perturbations even with large particle sizes.

## 5 Real-world experiment

### 5.1 Experimental settings

Here the OSSE is extended to the real world by replacing the simulated observations with the real observations. The sensitivity results in the previous section showed that the settings used for the TEST experiment provided stable filter performance; therefore, we follow the TEST experiment here with the moderate initial and resampling perturbation sizes and with 8000 particles.

Since the OSSE used the actual climate forcing in 2004 to 2007, we used the quality-controlled MODIS LAI product of MCD15A3 for those years with flagged as "good quality", "Terra or Aqua", "detectors apparently fine for up to 50 %", "significant clouds not present", and "main method used with or without saturation". We took the median of the LAI observations in the 10-km radius from the study site (62º 15' 18" N, 129º 14' 29" E). There are a number of missing data in the quality-controlled MODIS data. Therefore, if the number of the data in the 10-km radius is less than 300, we set these data as the missing data for DA. Since the MODIS data resolution is 1 km, the 10-km radius area contains about 314 data. The observation error standard deviations are assigned to each LAI datum in the original MODIS product (Knyazikhin et al., 1999). We rely on the estimate of the observation error standard deviations, and took the median of the error standard deviations in the same way as getting the LAI data. The observation error standard deviation is used in the particle filter when computing the likelihood function (Eq. 1).

The model-simulated NEE was validated with the field observation data at this AsiaFlux site (Ohta et al., 2001; 2008; 2014). The data was quality controlled by the steady-state test as indicated by the quality flag 0. Although the model simulates daily-average NEE, the field observation data represent instantaneous NEE every 30 minutes. The observation data are missing frequently, and it is not trivial to derive daily averages. Therefore, the raw data are compared with the DA results directly. This allows only a rough verification about whether or not the simulated NEE is in a reasonable range, but this is the only possible verification with an independent source.

### 5.2 Results

Figures 7, 8, and 9 show similar time series to Figs. 3, 4, and 5, respectively, but with the real MODIS LAI observations. Although the particles of the NODA experiments are widely spread, DA makes the particles much narrower (right) for all variables and parameters. With DA, the median of LAI is very close to the observations, within the range of the observation error standard deviations (Fig. 7 a). The grass and tree LAIs are estimated separately (Fig. 7 b, c), but there is no direct observation or other verification truth to compare with. This is similar to the model parameters (Fig. 8) and other model variables (Fig. 9) except for NEE, for which direct field observation data are available. As in the OSSE results, the range of uncertainties for NEE is reduced significantly by DA (Fig. 9 c). Since the field observations are made instantaneously every 30 minutes, the observation values (red) appear to have a wider range. However, the SEIB-DGVM simulates only daily-

average NEE, and it is not straightforward to compare the outputs from SEIB-DGVM with the field observations. We still find that the median of NEE becomes closer to the observations, particularly near the dormancy period. The simulated NEE generally stays within the reasonable range compared with the field observations. In general, the particle filter shows promising results with the real MODIS LAI data.

## 6 Conclusion

We assimilated the satellite-based MODIS LAI data using a non-Gaussian ensemble DA system with the SEIB-DGVM based on the SIR particle filter approach. To the best of the authors' knowledge, this is the first study to assimilate the fine time-scale satellite data with an *individual-based* DGVM. We found that DA performed generally well both for OSSE and real-world experiment. Although we assimilated only LAI as a whole, the tree and grass LAIs were estimated separately. This suggests that the satellite-based DA reduce the uncertainties in the initial vegetation structure of the *individual-based* DGVM toward the simulation of the future vegetation change. Another notable ~~results include~~ result includes that the model parameters of the *individual-based* DGVM were estimated successfully, and that the uncertainties in the unobserved model variables relevant to carbon cycle and vegetation states were also reduced significantly. Similarly to the previous studies with a *static* vegetation model (Stöckli et al., 2011) and a *non-individual-based* DGVM (Demarty et al., 2007), the results in the present study also suggest that LAI be the key to DA for phenology and carbon dynamics.

Generally, particle filters do not work well in high-dimensional problems (e.g., Bickel et al., 2008; Snyder et al., 2008; 2012; 2015). The SEIB-DGVM has several thousand state variables, but we applied random perturbations to only four model parameters in the particle filter. The four model parameters, i.e., Pmax and Dor for tree and grass, control the leaf season and photosynthesis rate of the forest as a whole. Therefore, the effective degrees of freedom of the estimation problem would be substantially lower than the number of variables of the SEIB-DGVM. This may be why the particle filer worked well in this study.

Additional sensitivity experiments revealed general robustness but some sensitivities to the nature run, initial and resampling perturbation sizes, and particle size, particularly for biomass, which tends to show particle degeneracy. When resampling, the random perturbations were applied only to Pmax and Dor, not to model state variables. This contributes to reduce the variety of vegetation structures such as tree densities and tree heights due to the frequent DA every four days (not shown). This tends to cause particle degeneracy for biomass even with large particle sizes when the resampling perturbation size is small (Table 5). When the resampling perturbation size is relatively large, degeneracy of the vegetation structure is mitigated to some extent. Therefore, in this study, we tuned the resampling perturbation sizes to avoid the filter collapse for biomass, and found that the "moderate" perturbation size with 8000 particles is a reasonable choice. However, the moderate perturbation size may be large for variables other than biomass, and this may be why the estimated parameters show occasional jumps. Adding resampling perturbations to other variables in addition to Pmax and Dor would be better. Also, since resampling

perturbations affect the particle spread strongly, the DA technique does not necessarily provide accurate estimates of the errors. In the future study, we will explore more effective resampling methods to avoid the filter collapse for biomass and to represent error estimates more accurately.

As a potential limitation, it is important to note that we have made strong assumptions in OSSE. For example, the only source of model imperfections was the model parameter uncertainties of the four parameters. It was also assumed that the observation error statistics were perfectly known. These conditions would have never been met in the real-world experiment.

As the first step, this study focused on the methodological development of the data assimilation system with SEIB-DGVM and estimated only four parameters of two PFTs using LAI observations at a single location. As a next step, more parameters and distributions of more diverse PFTs should be considered at different locations. Local-scale experiments can be performed in parallel for different locations since the satellite-based LAI observations are available globally. The simulation with the initial states and parameter sets obtained from the SEIB-DGVM-based DA system would be expected to improve the estimates of the carbon cycle changes over the globe.

**Appendix: List of modifications to SEIB-DGVM ver.2.71.**

without asterisk: modifications in this study, * Sato, personal communications, 2014, ** Sato et., al 2016

| | |
|---|---|
| **Modification to main.f90** | |
| SUBROUTINE main_loop | |
|   Intialize variables: parameters (Pmax for tree and grass, Dor for tree and grass) are read here. | |
|   Wild fire subroutines: Fire function was excluded. | |
| **Modification to metabolic.f90** | |
| SUBROUTINE photosynthesis_condition: | |
|   ce_water (no dimension, the minimum value): limitation on photosynthesis via soil water: | |
|   x = min( 1.0 , max(0.001, stat_water(p)) ) $\rightarrow$ x = min( 1.0 , max(0.1, stat_water(p)) ) | |
| SUBROUTINE leaf_season: | |
|   Days_leaf_shed (days): day length required for full leaf drop $\rightarrow$ from 14 to 30 | |
|   Days_release_larch (days): days required for full release of stock energy for larch $\rightarrow$ from 7 to 60 | |
|   Checker (Foliage $\rightarrow$ Domance) | |
|     case (1) : if (x < 7.0) flag(p)=.true. $\rightarrow$ if (doy >= Dor_f) flag(p)=.true. (doy: day of the year, Dor_f: Dor for tree) | |
|     case (5:6) : if (y > 0.01) flag(p)=.false. $\rightarrow$ if (doy >= Dor_g) flag(p)=.true. (doy: day of the year, Dor_g: Dor for grass) | |
|     if (dfl_leaf_onset (p) $\rightarrow$ Days_foliation_min) flag(p)=.false. $\rightarrow$ comment out | |
|   Checker (Dormance $\rightarrow$ Foliage) | |
|     case (1) : if ( x >= 65.0 ) flag(p) =.true. $\rightarrow$ if ( doy>=110) flag(p) = .true (doy: day of the year) | |
|     case (5:6) : if (y <= 0.01) flag(p) = .false. $\rightarrow$ if ( doy>=110) flag(p) = .true (doy: day of the year) | |
|   Gradual release of stock energy : (for bug fix) | |
|     IF (dfl_leaf_onset (p) >= day_length_release) cycle $\rightarrow$IF (dfl_leaf_onset(p) >= (day_length_release-1)) cycle | |
| SUBROUTINE maintenance_resp: | |
|   Herbaceous PFTs  Source 1: (for bug fix) *) | |
|     mass_combust = mass_combust + mass_required $\rightarrow$ mass_combust = mass_combust + mass_required * x | |
|     npp_(p) = npp_(p) - mass_required $\rightarrow$ npp_(p) = npp_(p) - mass_required * x | |
| SUBROUTINE growth_wood: | |
|   Delay_from_foliation (days): delay of stem growth and reproduction process after foliation $\rightarrow$ from 21 to 0 * | |
| **Modification to parameter.txt** | |
| TO_f (times / year): turn over time for foliage (grass) | from 0.50 to 3.19 ** |
| TO_r (times / year): turn over time for root (tree). We set the same value as the other boreal tree PFTs. | from 0.16 to 0.42 |
| ALM1 ($m^2$ / m):Allometry index of LA vs dbh of sapwood (tree) | from 6000 to 0 ** |
| ALM3 (g dm / $m^3$): Allometry index of trunk mass (tree) | from 0 to 700000 ** |
| FR_ratio (g dm / g dm): ratio of leaf mass vs root mass (tree) | from 0.17 to 0.35 ** |
| FR_ratio (g dm / g dm): ratio of leaf mass vs root mass (grass) | from 0.33 to 0.10 * |
| SLA (one sided $m^2$ / g dm): specific leaf area (tree) | from 0.014 to 0.010 ** |
| SLA (one sided $m^2$ / g dm): specific leaf area (grass) | from 0.015 to 0.020 ** |
| Topt0 (°C ): optimum temperature (tree) | from 20.0 to 21.0 ** |
| Tmin (°C): minimum temperature (tree). We set the same value as the other boreal tree PFTs. | from 5.0 to -4.0 |
| Tmax (°C ): maximum temperature (tree) | from 35.0 to 38.0** |
| GS_b2 (no dimension): parameters of stomatal conductance (grass) | from 3.0 to 5.0 * |
| M1 (no dimension): asymptotic maximum mortality rate (tree) | from 0.003 to 0.001 ** |
| TC_min (°C ): minimum coldest month temperature for persisting (tree and grass) | from -1000.0 to -45.0 ** |
| GDD_min (5 °C_base): minimum degree-day sum for establishment (tree) | from 350 to 250 ** |
| Est_scenario: Scenario for establishment for tree. Only specified woody PFT was set to establish. | * |
| Est_pft_OnOff: Establish switch for tree. Only boreal deciduous needle leaved tree was set to establish. | * |

**Acknowledgements**

The authors thank Hisashi Sato, the main developer of SEIB-DGVM, for useful discussions. The authors also thank reviewers Matthias Morzfeld and Malaquias Pena Mendez for their careful reviews and constructive comments that helped improve the manuscript significantly. The source code of SEIB-DGVM and the climate forcing data are available at http://seib-dgvm.com/.

MODIS LAI product of MCD15A3 was retrieved from the online Data Pool, courtesy of the NASA Land Processes Distributed Active Archive Center (LP DAAC), USGS/Earth Resources Observation and Science (EROS) Center, Sioux Falls, South Dakota (https://lpdaac.usgs.gov/data_access/data_pool). Carbon flux data was retrieved from AsiaFlux database (https://db.cger.nies.go.jp/asiafluxdb/).

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

**Table 1**. Parameter settings for TEST, OSSE2 and OSSE3.

| OSSEs | Pmax for tree | Pmax for grass | Dor for tree | Dor for grass |
|-------|---------------|----------------|--------------|---------------|
| TEST  | 15            | 9              | 230          | 270           |
| OSSE2 | 20            | 12             | 220          | 260           |
| OSSE3 | 25            | 7              | 210          | 280           |

**Table 2**. Initial perturbation settings.

| Initial perturbation sizes | Pmax for tree | Pmax for grass | Dor for tree | Dor for grass |
|---|---|---|---|---|
| Small | [0, 20] | [0, 10] | [200, 250] | [250, 300] |
| Moderate | [0, 60] | [0, 15] | [200, 300] | [200, 300] |
| Large | [0, 120] | [0, 30] | [150, 350] | [150, 350] |

**Table 3**. Results for the sensitivity experiments on the initial perturbation size: (a) mean absolute error (MAE) and (b) the widths of the 1-99% quantiles, averaged over a year in 2007. Gray shades show the filter divergence. Bold letters show the TEST experiment (8000 particles with moderate initial perturbations).

(a)

| Particle sizes | Initial perturbation sizes | Pmax for tree | Pmax for grass | Dor for tree | Dor for grass | LAI (m²/m²) | Biomass (MgC/ha) | GPP (MgC/ha/day) | RE (MgC/ha/day) | NEE (MgC/ha/day) |
|---|---|---|---|---|---|---|---|---|---|---|
| 1000 | SMALL | 0.76 | 0.75 | 0.43 | 3.68 | 0.03 | 2.46 | 0.003 | 0.001 | 0.002 |
| | MODERATE | 3.49 | 1.59 | 0.64 | 0.85 | 0.02 | 4.75 | 0.003 | 0.001 | 0.003 |
| | LARGE | 9.18 | 0.59 | 33.41 | 35.70 | 0.10 | 0.21 | 0.007 | 0.002 | 0.006 |
| 2000 | SMALL | 0.66 | 0.52 | 0.51 | 3.83 | 0.03 | 1.36 | 0.003 | 0.001 | 0.002 |
| | MODERATE | 3.49 | 2.05 | 0.71 | 3.11 | 0.04 | 3.29 | 0.003 | 0.001 | 0.002 |
| | LARGE | 3.30 | 1.21 | 1.01 | 59.52 | 0.10 | 9.22 | 0.003 | 0.002 | 0.003 |
| 4000 | SMALL | 0.97 | 0.26 | 0.79 | 2.38 | 0.03 | 1.16 | 0.002 | 0.001 | 0.002 |
| | MODERATE | 3.55 | 1.20 | 0.63 | 3.02 | 0.03 | 2.80 | 0.003 | 0.002 | 0.003 |
| | LARGE | 3.46 | 0.69 | 1.23 | 2.46 | 0.03 | 0.39 | 0.003 | 0.002 | 0.003 |
| 8000 | SMALL | 0.69 | 0.30 | 0.78 | 2.30 | 0.03 | 0.24 | 0.002 | 0.001 | 0.002 |
| | **MODERATE** | **2.88** | **1.10** | **0.73** | **7.83** | **0.03** | **3.72** | **0.003** | **0.001** | **0.003** |
| | LARGE | 3.36 | 0.99 | 1.18 | 3.20 | 0.04 | 1.60 | 0.003 | 0.001 | 0.003 |
| 16000 | SMALL | 2.92 | 1.01 | 0.72 | 6.91 | 0.03 | 0.80 | 0.003 | 0.001 | 0.003 |
| | MODERATE | 3.22 | 1.13 | 0.65 | 2.16 | 0.03 | 3.08 | 0.003 | 0.001 | 0.003 |
| | LARGE | 3.02 | 1.03 | 0.39 | 6.26 | 0.03 | 1.72 | 0.003 | 0.001 | 0.002 |

(b)

| Particle sizes | Initial perturbation sizes | Pmax for tree | Pmax for grass | Dor for tree | Dor for grass | LAI (m²/m²) | Biomass (MgC/ha) | GPP (MgC/ha/day) | RE (MgC/ha/day) | NEE (MgC/ha/day) |
|---|---|---|---|---|---|---|---|---|---|---|
| 1000 | SMALL | 14.93 | 5.10 | 23.44 | 28.26 | 0.17 | 4.63 | 0.013 | 0.005 | 0.012 |
| | MODERATE | 24.18 | 8.02 | 23.60 | 26.80 | 0.18 | 0.42 | 0.017 | 0.006 | 0.014 |
| | LARGE | 17.92 | 7.47 | 19.61 | 26.97 | 0.11 | 0.29 | 0.010 | 0.005 | 0.009 |
| 2000 | SMALL | 16.68 | 5.36 | 27.77 | 30.34 | 0.18 | 0.93 | 0.014 | 0.005 | 0.012 |
| | MODERATE | 24.08 | 9.05 | 26.24 | 29.50 | 0.20 | 0.63 | 0.018 | 0.007 | 0.015 |
| | LARGE | 27.30 | 9.23 | 25.97 | 54.89 | 0.24 | 0.62 | 0.017 | 0.006 | 0.014 |
| 4000 | SMALL | 15.72 | 4.60 | 24.18 | 28.76 | 0.14 | 4.89 | 0.012 | 0.005 | 0.011 |
| | MODERATE | 27.11 | 8.62 | 27.73 | 28.68 | 0.20 | 0.70 | 0.018 | 0.007 | 0.015 |
| | LARGE | 27.07 | 8.12 | 27.59 | 29.29 | 0.19 | 0.60 | 0.016 | 0.006 | 0.014 |
| 8000 | SMALL | 16.02 | 4.50 | 25.38 | 30.14 | 0.15 | 7.27 | 0.012 | 0.005 | 0.011 |
| | **MODERATE** | **28.32** | **9.29** | **26.23** | **33.99** | **0.20** | **11.40** | **0.017** | **0.008** | **0.015** |
| | LARGE | 27.47 | 9.37 | 26.60 | 31.75 | 0.21 | 0.71 | 0.017 | 0.007 | 0.014 |
| 16000 | SMALL | 27.66 | 9.28 | 27.18 | 48.79 | 0.22 | 8.44 | 0.017 | 0.007 | 0.015 |
| | MODERATE | 28.47 | 8.85 | 27.86 | 31.91 | 0.21 | 6.53 | 0.017 | 0.008 | 0.015 |
| | LARGE | 28.76 | 8.93 | 25.77 | 47.88 | 0.21 | 2.08 | 0.017 | 0.006 | 0.014 |

**Table 4**. Resampling perturbation settings.

| Resampling perturbation sizes | Pmax for tree | Pmax for grass | Dor for tree | Dor for grass |
|---|---|---|---|---|
| Small | [-2, 2] | [-0.5, 0.5] | [-2, 2] | [-2, 2] |
| Moderate | [-4, 4] | [-1, 1] | [-4, 4] | [-4, 4] |
| Large | [-8, 8] | [-2, 2] | [-8, 8] | [-8, 8] |

**Table 5**. Similar to Table 3, but for the sensitivity experiments on the resampling perturbation sizes.

(a)

| Particle sizes | Resampling perturbation sizes | Pmax for tree | Pmax for grass | Dor for tree | Dor for grass | LAI (m²/m²) | Biomass (MgC/ha) | GPP (MgC/ha/day) | RE (MgC/ha/day) | NEE (MgC/ha/day) |
|---|---|---|---|---|---|---|---|---|---|---|
| 500 | SMALL | 2.59 | 1.68 | 1.74 | 15.74 | 0.08 | 1.47 | 0.003 | 0.001 | 0.002 |
| | MODERATE | 2.28 | 0.66 | 1.34 | 5.57 | 0.03 | 1.43 | 0.003 | 0.001 | 0.003 |
| | LARGE | 10.13 | 1.25 | 2.55 | 5.29 | 0.04 | 0.35 | 0.004 | 0.002 | 0.003 |
| 1000 | SMALL | 0.80 | 0.58 | 1.09 | 1.41 | 0.02 | 4.62 | 0.002 | 0.002 | 0.003 |
| | MODERATE | 3.49 | 1.59 | 0.64 | 0.85 | 0.02 | 4.75 | 0.003 | 0.001 | 0.003 |
| | LARGE | 5.00 | 2.17 | 2.78 | 8.56 | 0.04 | 4.78 | 0.004 | 0.002 | 0.004 |
| 2000 | SMALL | 1.84 | 0.42 | 0.47 | 7.94 | 0.02 | 0.59 | 0.002 | 0.001 | 0.002 |
| | MODERATE | 3.49 | 2.05 | 0.71 | 3.11 | 0.04 | 3.29 | 0.003 | 0.001 | 0.002 |
| | LARGE | 6.84 | 1.26 | 2.60 | 4.26 | 0.05 | 0.79 | 0.004 | 0.002 | 0.003 |
| 4000 | SMALL | 1.41 | 0.67 | 0.56 | 0.36 | 0.02 | 2.94 | 0.002 | 0.002 | 0.002 |
| | MODERATE | 3.55 | 1.20 | 0.63 | 3.02 | 0.03 | 2.80 | 0.003 | 0.002 | 0.003 |
| | LARGE | 6.74 | 1.17 | 1.66 | 6.94 | 0.05 | 2.15 | 0.004 | 0.002 | 0.003 |
| 8000 | SMALL | 1.23 | 0.37 | 0.61 | 0.71 | 0.02 | 3.20 | 0.002 | 0.001 | 0.002 |
| | **MODERATE** | **2.88** | **1.10** | **0.73** | **7.83** | **0.03** | **3.72** | **0.003** | **0.001** | **0.003** |
| | LARGE | 7.78 | 1.56 | 1.26 | 7.46 | 0.05 | 3.44 | 0.004 | 0.002 | 0.003 |
| 16000 | SMALL | 1.46 | 0.45 | 0.77 | 1.70 | 0.02 | 3.15 | 0.002 | 0.001 | 0.002 |
| | MODERATE | 3.22 | 1.13 | 0.65 | 2.16 | 0.03 | 3.08 | 0.003 | 0.001 | 0.003 |
| | LARGE | 5.67 | 1.71 | 1.50 | 6.07 | 0.05 | 1.36 | 0.004 | 0.002 | 0.003 |

(b)

| Particle sizes | Resampling perturbation sizes | Pmax for tree | Pmax for grass | Dor for tree | Dor for grass | LAI (m²/m²) | Biomass (MgC/ha) | GPP (MgC/ha/day) | RE (MgC/ha/day) | NEE (MgC/ha/day) |
|---|---|---|---|---|---|---|---|---|---|---|
| 500 | SMALL | 12.41 | 3.98 | 11.29 | 18.21 | 0.13 | 0.37 | 0.012 | 0.004 | 0.010 |
| | MODERATE | 25.82 | 7.47 | 23.68 | 31.83 | 0.18 | 0.44 | 0.015 | 0.006 | 0.013 |
| | LARGE | 47.51 | 11.38 | 52.32 | 40.18 | 0.24 | 0.70 | 0.022 | 0.008 | 0.018 |
| 1000 | SMALL | 14.31 | 4.17 | 13.85 | 16.57 | 0.14 | 0.46 | 0.010 | 0.004 | 0.009 |
| | MODERATE | 24.18 | 8.02 | 23.60 | 26.80 | 0.18 | 0.42 | 0.017 | 0.006 | 0.014 |
| | LARGE | 44.68 | 11.57 | 53.50 | 45.94 | 0.24 | 0.60 | 0.022 | 0.009 | 0.018 |
| 2000 | SMALL | 14.10 | 4.40 | 12.69 | 31.61 | 0.16 | 0.46 | 0.010 | 0.004 | 0.009 |
| | MODERATE | 24.08 | 9.05 | 26.24 | 29.50 | 0.20 | 0.63 | 0.018 | 0.007 | 0.015 |
| | LARGE | 46.88 | 12.23 | 49.80 | 46.06 | 0.25 | 0.82 | 0.023 | 0.009 | 0.018 |
| 4000 | SMALL | 14.27 | 5.03 | 13.62 | 15.53 | 0.14 | 0.51 | 0.011 | 0.004 | 0.009 |
| | MODERATE | 27.11 | 8.62 | 27.73 | 28.68 | 0.20 | 0.70 | 0.018 | 0.007 | 0.015 |
| | LARGE | 43.99 | 12.93 | 50.11 | 45.44 | 0.25 | 6.28 | 0.023 | 0.010 | 0.019 |
| 8000 | SMALL | 15.30 | 5.07 | 13.78 | 16.72 | 0.15 | 0.67 | 0.011 | 0.004 | 0.009 |
| | **MODERATE** | **28.32** | **9.29** | **26.23** | **33.99** | **0.20** | **11.4** | **0.017** | **0.008** | **0.015** |
| | LARGE | 45.92 | 12.70 | 53.92 | 44.86 | 0.25 | 8.39 | 0.024 | 0.010 | 0.020 |
| 16000 | SMALL | 15.36 | 5.33 | 13.70 | 23.18 | 0.15 | 2.13 | 0.011 | 0.004 | 0.010 |
| | MODERATE | 28.47 | 8.85 | 27.86 | 31.91 | 0.21 | 6.53 | 0.017 | 0.008 | 0.015 |
| | LARGE | 44.05 | 12.27 | 52.21 | 46.10 | 0.25 | 7.76 | 0.024 | 0.010 | 0.019 |

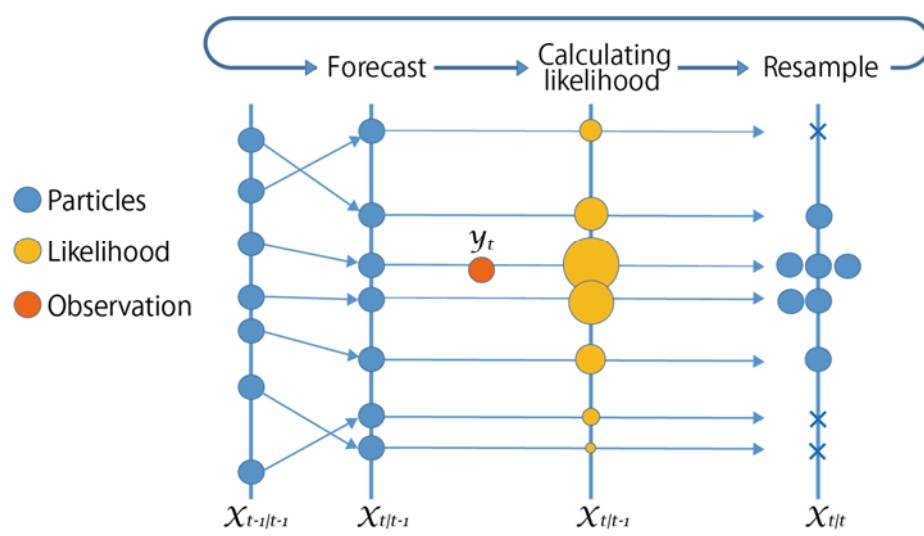

**Figure 1.** Schematic showing the SIR particle filter method. The size of the circles corresponds to the assigned probability.

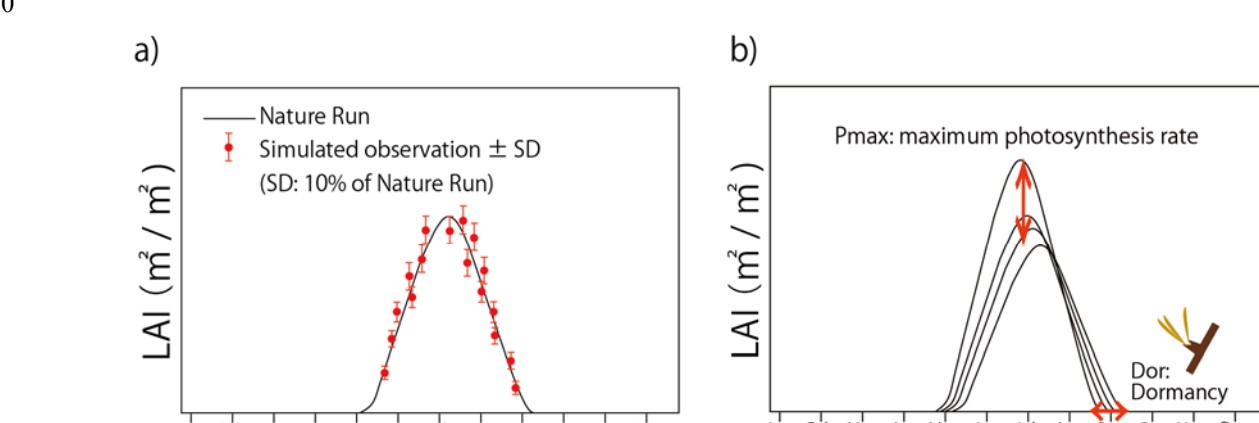

**Figure 2.** Schematic illustrations of the nature run, observations, and model parameter sensitivities. a) Time series of LAI (m²
m$^{-2}$) for the nature run (black), simulated observations (red dots) and their error standard deviations (SD, red error bars). b)
Time series of LAI with different Pmax and Dor values. The perturbed parameters (Pmax and Dor for tree and grass) cause
differences between the particles.

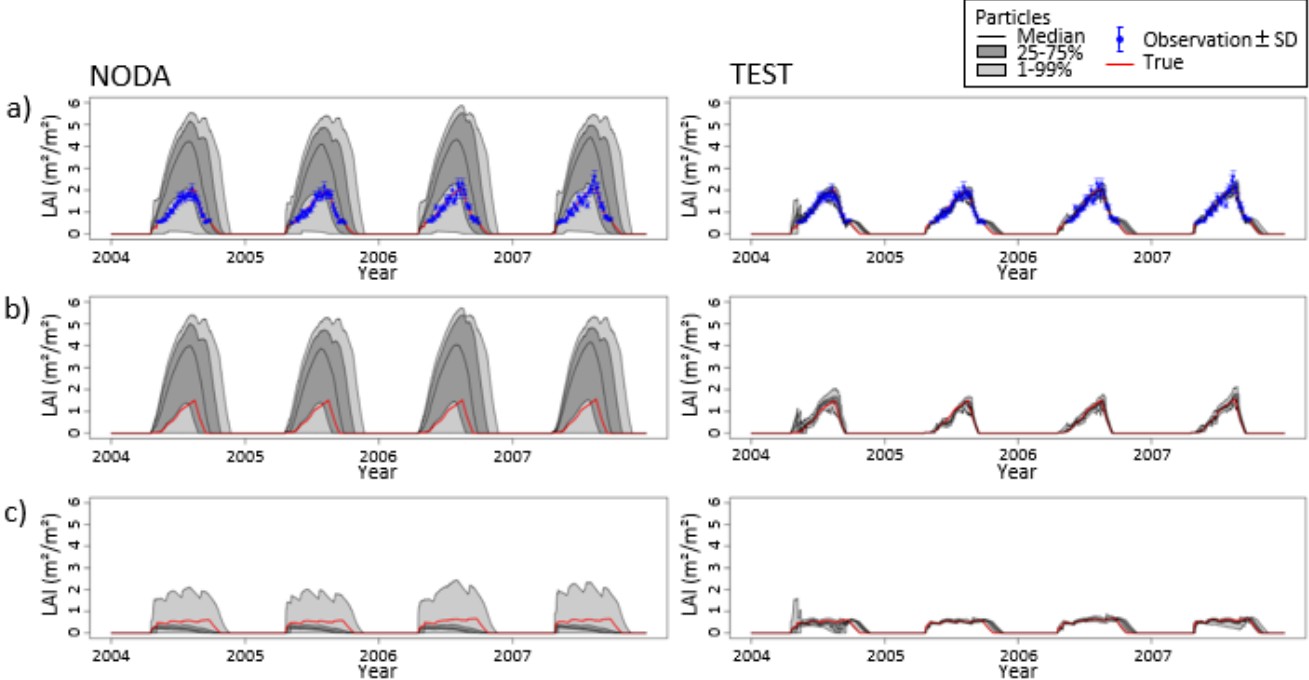

**Figure 3.** Time series of LAI for a) tree + grass, b) tree, and c) grass for NODA (left) and TEST (right). Dark and light gray
areas indicate the quartiles and 1-99 % quantiles of the particles as shown in legend. Thick black curves indicate the medians.
Blue dots with error bars indicate the observations and their error standard deviations, and red lines indicate the nature run.

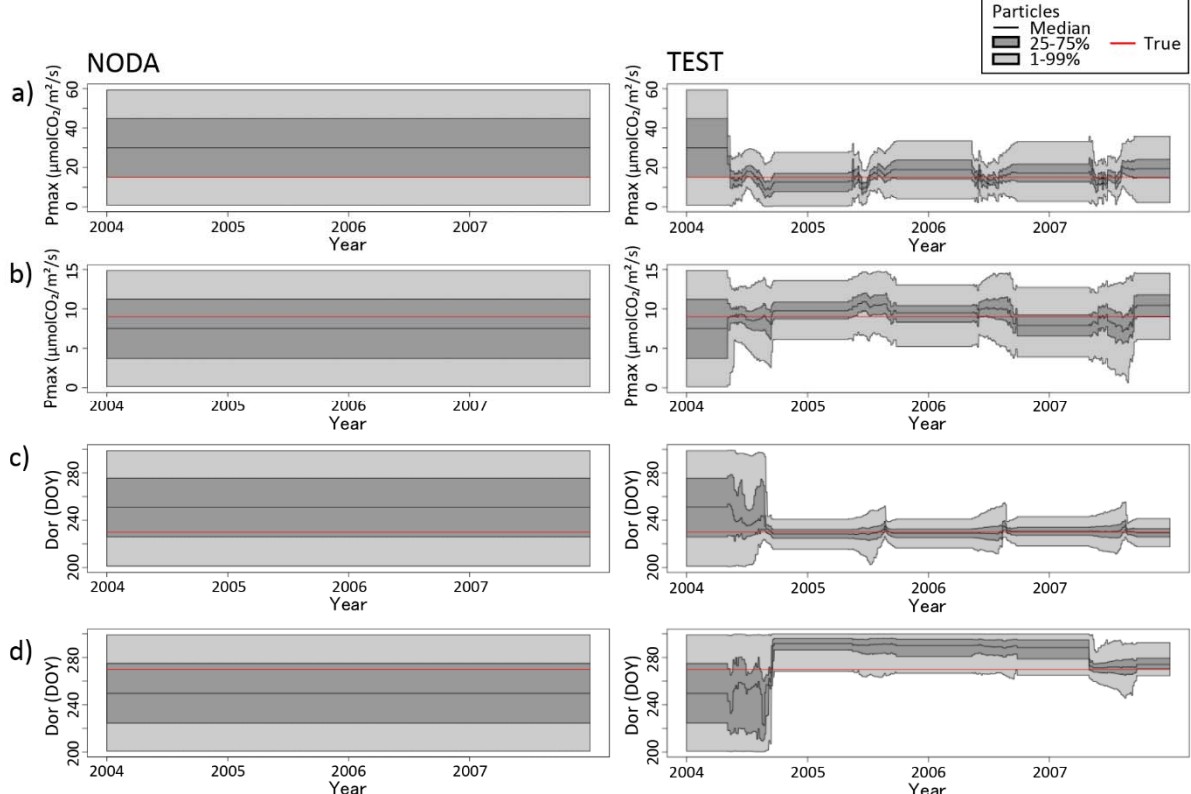

**Figure 4.** Similar to Fig. 3., but for the model parameters: a) Pmax for tree, b) Pmax for grass, c) Dor for tree, d) Dor for grass. There is no observation for these parameters.

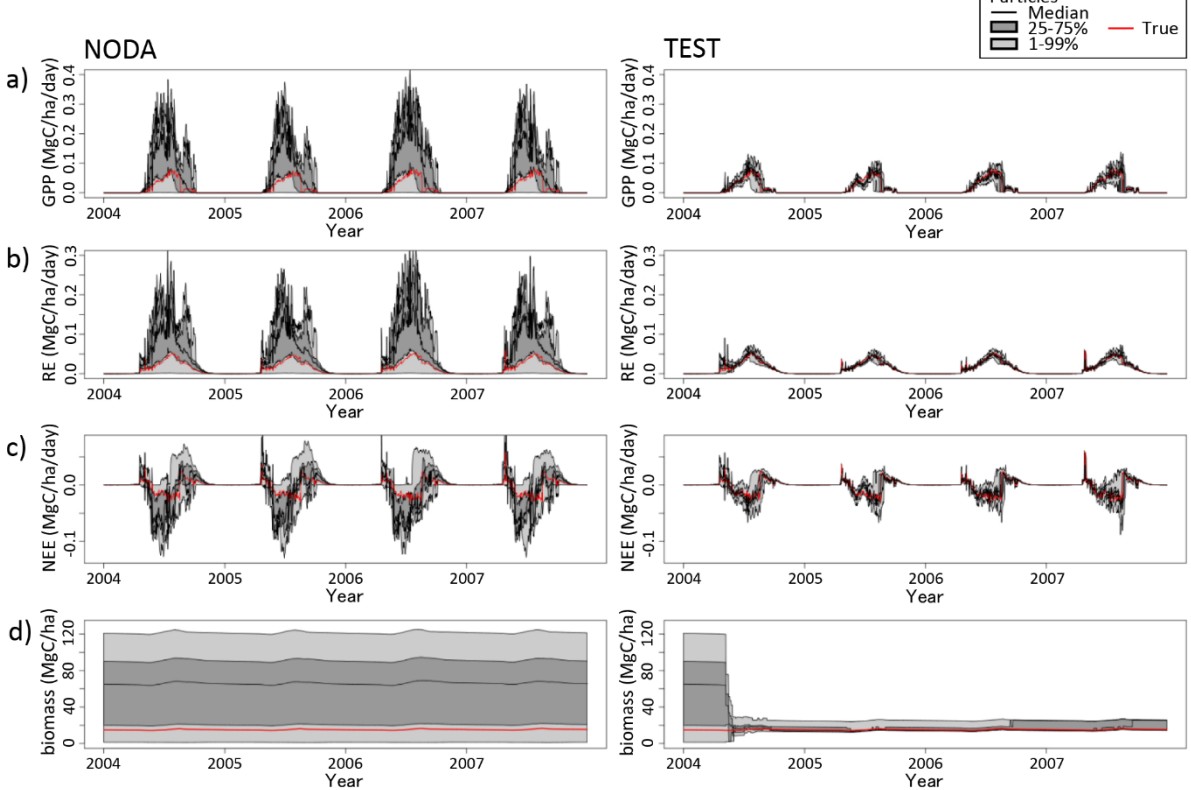

**Figure 5.** Similar to Figs. 3 and 4, but for unobserved model variables: a) GPP, b) RE, c) NEE, and d) biomass.

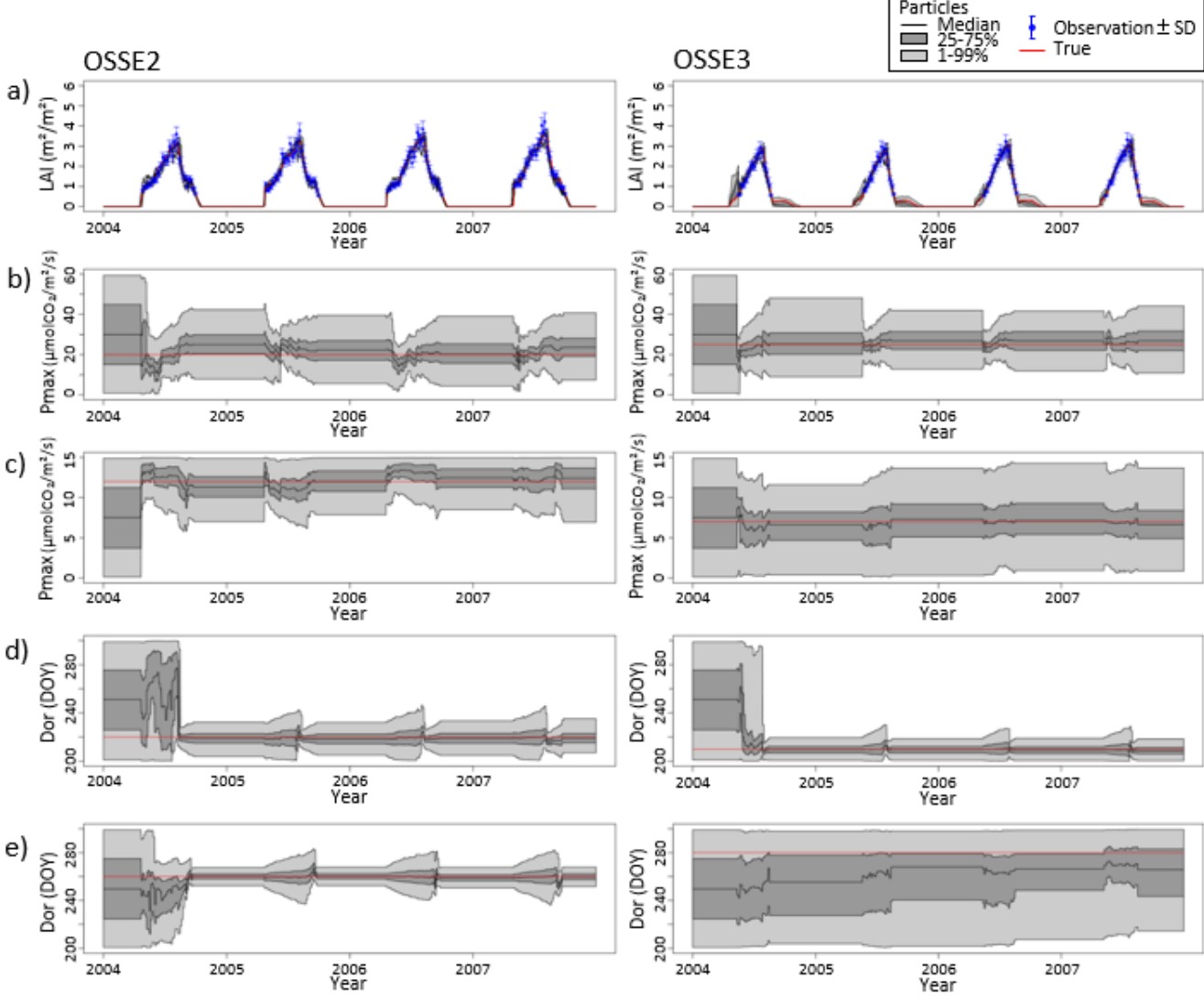

**Figure 6.** Similar to Figs. 3 and 4, but for OSSE2 (left) and OSSE3 (right). a) Time series of LAI for tree + grass, b) Pmax for tree, c) Pmax for grass, d) Dor for tree, e) Dor for grass.

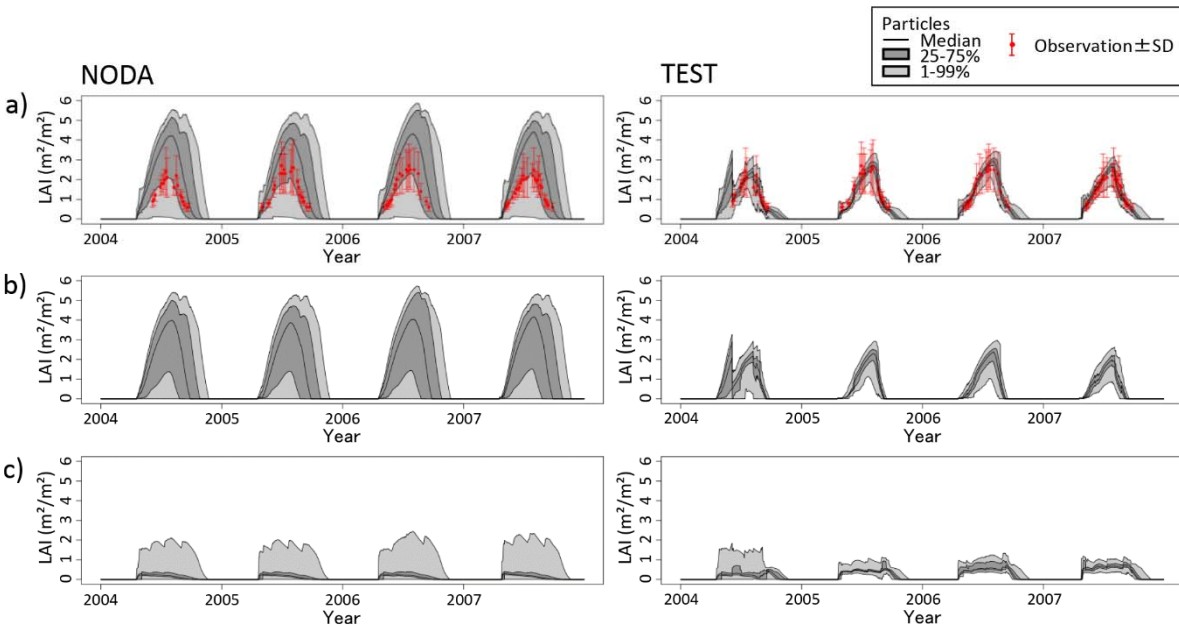

**Figure 7.** Similar to Fig. 3, showing LAI for a) tree + grass, b) tree, and c) grass, but for the real-world experiment. Red dots with error bars indicate the observations and their error standard deviations.

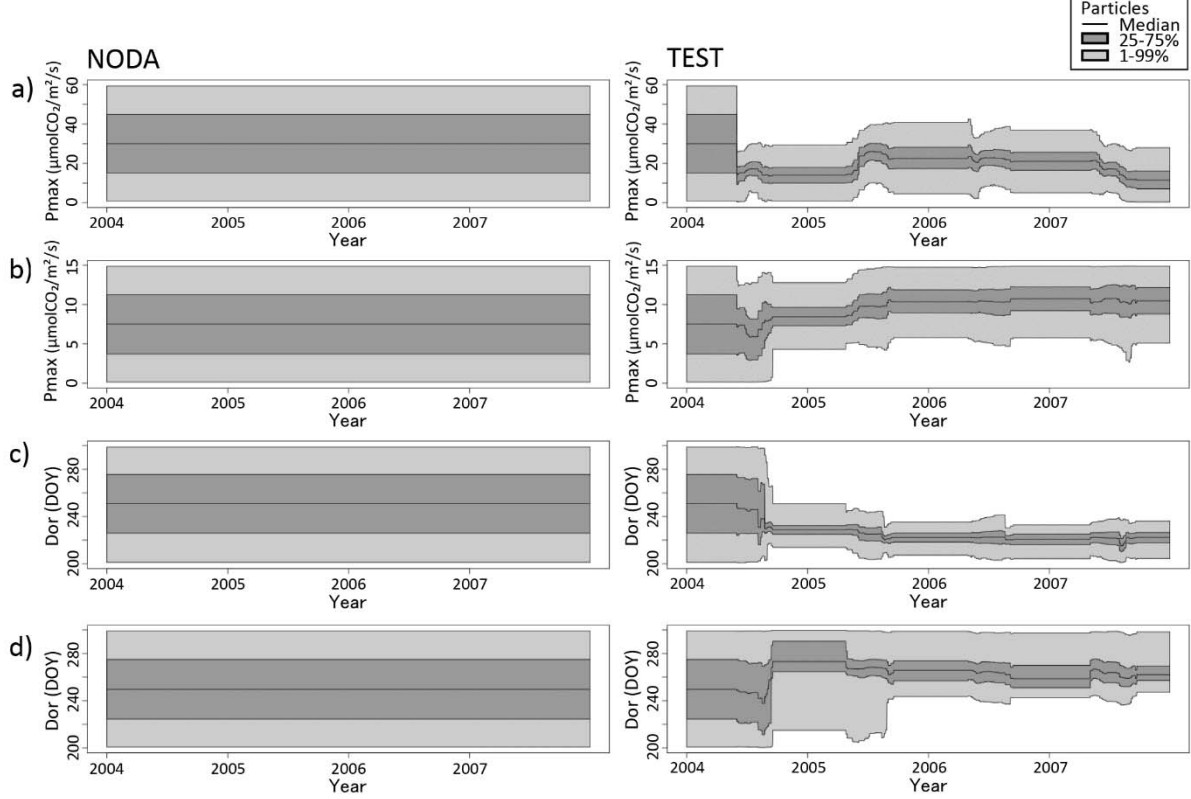

5  **Figure 8.** Similar to Fig. 4, showing the model parameters: a) Pmax for tree, b) Pmax for grass, c) Dor for tree, d) Dor for grass, but for the real-world experiment.

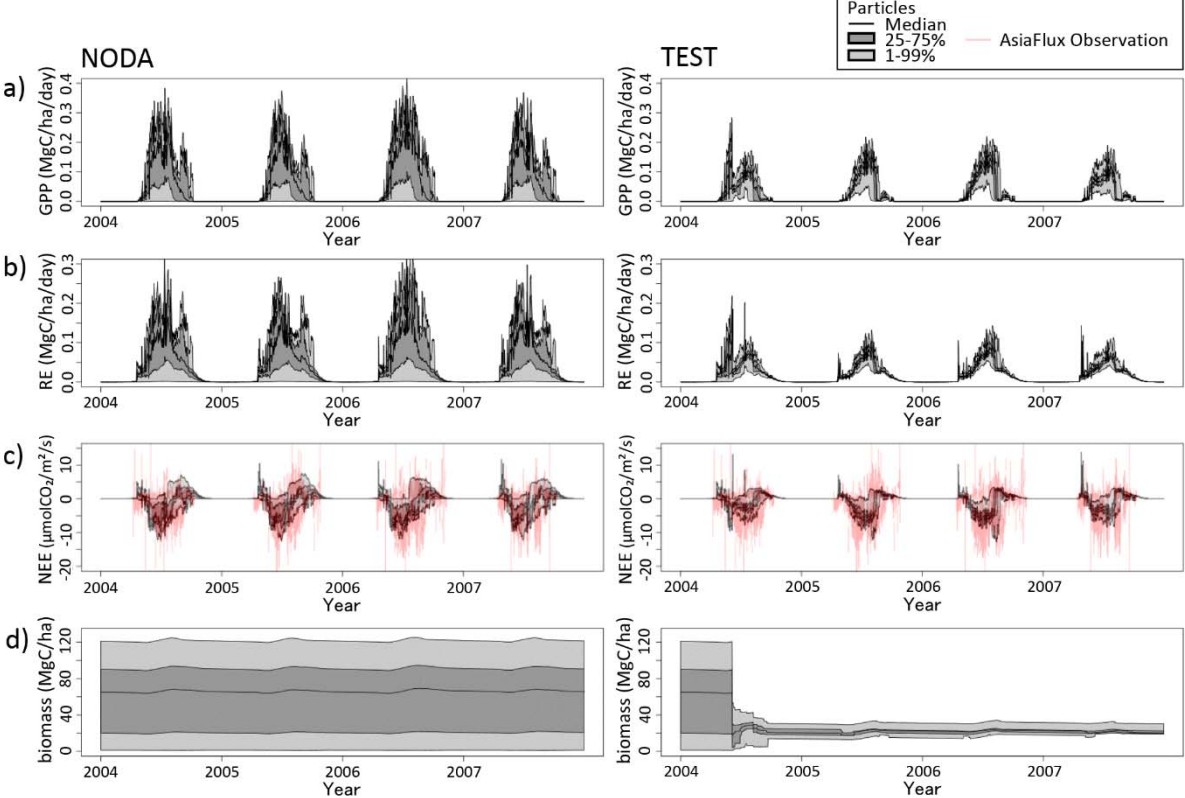

**Figure 9.** Similar to Fig. 5, showing the unobserved model variables: a) GPP, b) RE, c) NEE, d) biomass, but for the real-world experiment. Red lines indicate the direct field observations made instantaneously every 30 minutes at the AsiaFlux site, while the model simulates only the daily averages.