# Peer review of "Non-Gaussian data assimilation of satellite-based Leaf Area Index observations with an individual-based dynamic global vegetation model"

_Nonlinear Processes in Geophysics, 2016_

## Referee Comment (RC1) · M. Morzfeld (Referee) · 1 Jul 2016

General comments: The paper presents a description of the application of a particle filter to assimilate leaf are index observations into a global vegetation model. The paper is concise, and presents only the necessary information, no long background/review section is provided. I find the numerical experiments and results convincing (pending specific comments below).

My main concern is that this paper might not be suitable for Nonlinear Processes in Geophysics. The reason is that the journal emphasizes new methods, applied to realistic problems. The paper simply presents an "old" method applied to a new problem. I find it interesting to read that a particle filter can solve an important and "real" data assimilation, however the general NPG readership might get bored. The authors should decide wether NPG is the best journal to reach the audience they want to reach. This is also reflected by the references, only few of which are to articles in journals similar to NPG. I suspect that this paper would also make a fine contribution in a journal that is more focused on, e.g., Earth system modeling. The authors may want to consider going that route.

Specific comments:

(1) I wonder if there is any sensitivity to how repeated particles are perturbed after re-sampling. The authors chose a random perturbation, but miss to motivate their choice. I think the paper should contain numerical experiments where it is shown that either the method is robust to (small) changes in how repeated particles are perturbed, or it should be reported how the perturbations influence the results.

(2) The number of particle used is typically important for the results one obtains with a particle filter. Indeed, much of the meteorological literature says that the number of particles required is excessive. To address this issue, I would suggest to run more numerical experiments with a varying number of particles. One can then compute, e.g., means and variances, and check that the method has converged when, e.g., 8000 particles are used. Specifically, I suggest experiments with 4000, 8000 and perhaps 16000 particles (if possible).

(3) I wonder what happens when the data assimilation is initialized with a "smaller" initial uncertainty. The authors define intervals for the parameters, but do not mention how they came up with these intervals. It would be interesting to see what happens when these intervals are shortened or widened. In particular, the particle filter has no mechanism to bring the parameters to values that are not contained within the initial set. This could make things difficult for the "real life" application. Again, I suggest to investigate this issue with more numerical experiments.

(4) In figs.4 (right column), 5d (right column), 7 (right column), and 8d (right column): it

seems that the data assimilation only impacts the parameter estimates for parts of the year, however data are assimilated every 4 days. The authors miss to provide a clear explanation of why that is the case.

(5) I would remove all NODA figures, as they do not really carry information. It is clear that when no data assimilation is used, no parameter is changed.

Technical corrections:

I find the use of "newly" in the first sentence of the abstract a bit unusual. I would suggest to re-formulate this sentence. The sentence also appears again later on (p.2. line 6, p.6 line 27), and there it should also be changed.

––––––––––––––––––––––––––––––

---

## Referee Comment (RC2) · M. Pena (Referee) · 5 Jul 2016

General comments:

1. This paper applies a particle filter data assimilation scheme to assimilate MODIS Leaf Area Index (LAI) data into an explicit individual-plants dynamical global vegetation model. Results indicate that the scheme reduces the uncertainty of the LAI analyses as compared to random initialization. Furthermore, the technique appears to successfully estimate the model parameters that control separately LAI for the forest and for the grass types, out of whole LAI observations.

2. The content of the paper is relevant for Earth Systems and non-linear modeling. It

addresses one important aspect to increase models realism through the use of information contained in fine time-scale observed data.

3. The problem addressed in this study is challenging considering the nonlinearities in the dynamics of the vegetation, the multitude of interactive physical and biogeochemical processes taking place at the local and regional scales, and the fact that not all state variables are observed or retrieved by satellite.

4. The study is well executed to proof the concept with all the needed elements (calibrated model, quality controlled data, an optimized data assimilation scheme) and reduced (only a few geographical points) scope to make it successful.

Specific comments and questions:

1. While the description is succinct and easy to follow, it needs to make explicit major assumptions made and the problems one may encounter if they were to be relaxed.

2. Below is a list of questions that arose while reading the manuscript: 2.1: What modifications to the original vegetation model were made to adapt it to the DA scheme? Were these only the changes in parameter values we see in the appendix? 2.2: Are the field observations in the Siberia Yakutsk Larch forest site independent or were they included to create the climate forcing data 2001-2007 in the vegetation model? 2.3: Was the 2004-2007 period of MODIS 4-days frequency data continuous on the study site? Were there missing data? How does the missing data was handled? 2.4: Was the 8000 particles generated decided by computer capacity, or any other criteria? 2.5: Simulated observations (in the OSSE) versus real observations: How do they compare? Were the real observations also normally distributed? Were standard deviations of real observations about 10% as in the OSSE experiment? 2.6: Did you follow any particular rule to determine the perturbation size of Pmax and Dor? In the study you allow larger amplitude perturbations for forest than for grass types. The amplitude of Dor is relatively very small. 2.7: The manuscript indicates that perturbations of parameters are applied only to duplicated particles. Since the particle DA scheme eliminates

particles far away from observations (Fig. 1), that would mean that the range of the distribution of all the particles decreases after several cycles at least compared to the initial (uniform) distribution. Is this correct? Still, you do not report any issue with collapsing of the DA scheme when observations are outside the range of the distribution of particles. Can you please, elaborate more on this issue? 2.8: The NODA and the TEST experiments; Figure 3. How the 8000 particles are inserted at the initial conditions? Is this done every 4 days with a uniform distribution each time? The TEST experiment appears to reduce a big systematic error that appear during the growing months. Traditional DA schemes apply a bias-correction strategy of the First Guess prior to performing the analysis. Does this mean that particle DA also removes systematic errors? 2.9: TEST experiment; Figure 3a (forecast+grass). Please, explain the problem at the end of the fall months (circled in blue in the attached figure). Can this be attributed to neglecting observations when LAI<0.5 ? Will this be removed if observations are added there? 2.10: It is obscure to me how come the individual LAI of Forest and Grass are accurately estimated out of the whole LAI. Even when the whole LAI estimation is incorrect as in the periods in the blue circle in the attached figure. What mechanism or statistical assumption within the DA process makes the partitioning of LAI correct? Is this pure chance? 2.10: In the Real-World Experiment; there is no detail on the perturbation strategy, so I suppose it is the same as in the OSSE experiment. 2.11: The observation error standard deviation in the real case needs more explanation. What is the truth from which the error is estimated? Is this the in situ observation? Is this error an input in the DA scheme?

Technical corrections

Page 1. Abstract. ".. newly developed" should be ".. developed". You repeated that later on in the text.

Page 1. Abstract. ".., assuming the satellite-based LAI." This is an incomplete statement. You repeated this statement problem in the introduction, page 2, row 11. Maybe you meant to say "using" instead of "assuming".

Page 2. row 8. "straightforward" may be replaced by "numerically straightforward". In this context, it is not simple to go from local to global because spatial covariances become relevant.

Page 2. (last) row 31. "phase space stays the same" may be "phase space dimension stays the same"
* * *
[Figure]

**TEST**

Particles
Median
25-75%
1-99%

Observation ±SD
True

Fig. 1. Problem described in my comments are circled in blue

[Figure]

---

## Author Comment (AC2) · 30 Aug 2016

**Response to Referee #2 (Dr. Malaquias Pena Mendez)**

General comments:

1. This paper applies a particle filter data assimilation scheme to assimilate MODIS Leaf Area Index (LAI) data into an explicit individual-plants dynamical global vegetation model. Results indicate that the scheme reduces the uncertainty of the LAI analyses as compared to random initialization. Furthermore, the technique appears to successfully estimate the model parameters that control separately LAI for the forest and for the grass types, out of whole LAI observations.

2. The content of the paper is relevant for Earth Systems and non-linear modeling. It addresses one important aspect to increase models realism through the use of information contained in fine time-scale observed data.

3. The problem addressed in this study is challenging considering the nonlinearities in the dynamics of the vegetation, the multitude of interactive physical and biogeochemical processes taking place at the local and regional scales, and the fact that not all state variables are observed or retrieved by satellite.

4. The study is well executed to proof the concept with all the needed elements (calibrated model, quality controlled data, an optimized data assimilation scheme) and reduced (only a few geographical points) scope to make it successful.

Response: Thank you very much for the useful, constructive comments to improve the paper. We will revise the manuscript accordingly. Our point-by-point responses are shown in blue.

Specific comments and questions:

1. While the description is succinct and easy to follow, it needs to make explicit major assumptions made and the problems one may encounter if they were to be relaxed.

Response: Thank you very much for the suggestion. We made strong assumptions in the OSSE, and relaxed some in the real-world experiment. We still made strong assumptions in the limited area application. We will make these assumptions clear in the revised manuscript.

2. Below is a list of questions that arose while reading the manuscript:

2.1: What modifications to the original vegetation model were made to adapt it to the DA scheme? Were these only the changes in parameter values we see in the appendix?

Response: The model simulates daily states, but the original model outputs were only once per year. Daily outputs are needed for data assimilation every once in four days. Therefore, we modified the model code to output the daily states every 4 days. In addition, the original model code assumed running for many years continuously, and the initial seed for the random number generator was fixed. Since in this study we stopped the model every 4 days, and the same seed was repeated every time we started the model. Therefore, we modified the model code to randomly generate the seed for the random number generator every time when we initiate the model. These are the only modifications not shown in the paper, because we thought these were only minor technical modifications. To explicitly describe all necessary changes to the existing model code, we will include these in the appendix.

2.2: Are the field observations in the Siberia Yakutsk Larch forest site independent or were they included to create the climate forcing data 2001-2007 in the vegetation model?

Response: The climate forcing data were created using the NCEP/NCAR reanalysis data and CRU observation based data. The observed climate data at this site were not directly used in our experiments, but these data might be included in the NCEP/NCAR reanalysis data. It is not simple to find if the site observation data were reported through GTS and included in the NCEP/NCAR reanalysis. Therefore, we will include a sentence in the revised manuscript describing the possibility that the observations at the site may be used in the NCEP/NCAR reanalysis through GTS.

2.3: Was the 2004-2007 period of MODIS 4-days frequency data continuous on the study site? Were there missing data? How was the missing data handled?

Response: There are a number of missing data in the quality-controlled MODIS data. Therefore, as we have described in P5. Line 30-31, if the number of the quality controlled MODIS data in the 10-km radius contains less than 300 grid points, we set these data as the missing data. We will revise the manuscript to describe more explicitly about the missing data from the original quality-controlled MODIS data.

2.4: Was the 8000 particles generated decided by computer capacity, or any other criteria?

Response: In response to the other reviewer's comment #2, we performed additional experiments with different particle sizes. We will add a new section to show the sensitivity to

the particle size in the revised manuscript.

2.5: Simulated observations (in the OSSE) versus real observations: How do they compare? Were the real observations also normally distributed? Were standard deviations of real observations about 10% as in the OSSE experiment?

Response: We assumed the normal distribution for the real observation error. The error standard deviation is included in the MODIS dataset that we used (Knyazikhin et al., 1999). As already described in P.6 Lines 1-2, we used "the median of the error standard deviations" in the 10km radius. The standard deviations of the real observations are different from those of the OSSE, as indicated by Figs. 3-a and 6-a. We will explicitly describe about the differences of the observation error standard deviations between the simulated and real observations in the revised manuscript.

2.6: Did you follow any particular rule to determine the perturbation size of Pmax and Dor? In the study you allow larger amplitude perturbations for forest than for grass types. The amplitude of Dor is relatively very small.

Response: There are two perturbation settings for the model parameters: the initial perturbation sizes and the random perturbation sizes when resampling. We selected the initial perturbation sizes based on the ecological knowledge from the previous studies (Kolari et al., 2006; Zeng et al., 2011; Zhao et al., 2015; Takagi et al., 2015). The initial Pmax perturbation size for grass is 4 times smaller than that of forest. The initial Dor perturbation sizes for grass and forest are the same. The random perturbation sizes when resampling follow the initial perturbation sizes. We will add the references and explicit descriptions about the perturbation settings in the revised manuscript.

Following the comment of the other referee, we performed additional experiments with different random perturbation settings for the initial perturbation sizes and the random perturbation sizes when resampling. We will add a new section to show these results and to discuss the sensitivity to the perturbation settings in the revised manuscript.

2.7: The manuscript indicates that perturbations of parameters are applied only to duplicated particles. Since the particle DA scheme eliminates particles far away from observations (Fig. 1), that would mean that the range of the distribution of all the particles decreases after several cycles at least compared to the initial (uniform) distribution. Is this correct? Still, you do not

report any issue with collapsing of the DA scheme when observations are outside the range of the distribution of particles. Can you please, elaborate more on this issue?

Response: Yes, it is correct that the range of the distribution of all the particles decreases after several cycles. If we apply proper random perturbations to the duplicated particles, we can avoid filter collapse. However, our additional experiments showed filter collapse when the particle size is 1000. We will describe about the collapse in the new section on the sensitivity to the particle size and random perturbation size.

2.8: The NODA and the TEST experiments; Figure 3. How the 8000 particles are inserted at the initial conditions? Is this done every 4 days with a uniform distribution each time? The TEST experiment appears to reduce a big systematic error that appear during the growing months. Traditional DA schemes apply a bias-correction strategy of the First Guess prior to performing the analysis. Does this mean that particle DA also removes systematic errors?

Response: As already described in P.4 Line 28, "The 8000 particles at the end of the 103-year spin-up runs are used as the initial conditions for DA", and the NODA and TEST experiments start from the same initial 8000 particles. The 8000 particles continue to be the same until the first observation of LAI is assimilated. Since the LAI is observed only when greater than 0.5, the LAI observation exists only in the summer season. The systematic errors in NODA comes from the uncertain parameter settings. TEST can estimate parameters using observed LAI, and therefore, can reduce the systematic errors. This is different from the bias-correction strategy of the first guess. There is no explicit bias correction applied to the TEST experiment. So, we understand that this particular particle DA can reduce systematic errors by estimating the uncertain model parameters. We will discuss this point in the revised manuscript.

2.9: TEST experiment; Figure 3a (forecast+grass). Please, explain the problem at the end of the fall months (circled in blue in the attached figure). Can this be attributed to neglecting observations when LAI<0.5 ? Will this be removed if observations are added there?

Response: Yes, because we assume observation to be available only when LAI>0.5, it is difficult to estimate the LAI when observed LAI is 0.5 or smaller. We did not perform experiments with small LAI observations, because the MODIS data for the real-world experiment did not include LAI<=0.5 (Fig. 6). There are too few data with real MODIS LAI<=0.5, and our preprocessing assigns the missing value. We will add these discussions in the revised manuscript.

2.10: It is obscure to me how come the individual LAI of Forest and Grass are accurately estimated out of the whole LAI. Even when the whole LAI estimation is incorrect as in the periods in the blue circle in the attached figure. What mechanism or statistical assumption within the DA process makes the partitioning of LAI correct? Is this pure chance?

Response: Thank you for the comment, which initiated further analysis of the results that were not shown in the manuscript. As already described in P.5, Lines 21-22, "To investigate the sensitivity to the choice of the nature run, we performed similar OSSEs by replacing the nature run with other randomly-chosen parameter sets". We investigated these different OSSEs more carefully and found that the parameters for grass were estimated well when the nature run used a larger Pmax value for grass (Figure 1 left, as shown in the manuscript). However, in another OSSE (Figure 1 right), the nature run used a small Pmax value; the results showed that the parameters for grass showed significantly larger uncertainties (Fig. 1 c, e), while the parameters for forest were estimated well (Fig. 1 b, d). Larger Pmax values for grass produce more grass LAI, which can be observed with the observing threshold of LAI > 0.5 near the growing and falling periods (shown by the blue circles provided by the reviewer). With smaller Pmax values for grass, the small grass LAI cannot be observed directly, but the large LAI observations in the summer season predominantly suggest forest LAI. This would allow to estimate the forest parameters well, although the grass parameters showed larger uncertainties. We will include these results and discussions in the revised manuscript.

[Figure]

Figure 1. The results of the OSSE1 and OSSE3 with data assimilation. The particle sizes and the perturbation sizes are the same as the discussion paper. a) Time series of LAI for forest + grass, b) Pmax for forest, c) Pmax for grass, d) Dor for forest, e) Dor for grass. Dark and light gray areas indicate the quartiles and 1-99 % quantiles of the particles as shown in legend. Thick black curves indicate the medians. Red dots with error bars indicate the observations and their error standard deviations, and red and blue lines indicate the true values used for the nature runs.

2.10: In the Real-World Experiment; there is no detail on the perturbation strategy, so I suppose it is the same as in the OSSE experiment.

Response: We did not provide the details of the experimental settings. Yes, the perturbation strategy of the real-world experiment is same as that of the OSSE. We will add the descriptions in the revised manuscript.

2.11: The observation error standard deviation in the real case needs more explanation. What is the truth from which the error is estimated? Is this the in situ observation? Is this error an input in the DA scheme?

Response: As described in our previous response to the comment #2.5, P6 lines 1-2 reads "The observation error standard deviations are assigned to each LAI datum from the original source, and we took the median of the error standard deviations." We rely on the original MODIS data source about the estimate of the observation error standard deviation. We will revise this sentence about the observation error standard deviations to avoid potential misunderstanding.

Technical corrections
Page 1. Abstract. ".. newly developed" should be ".. developed". You repeated that later on in the text.

Response: We will correct it accordingly.

Page 1. Abstract. ".., assuming the satellite-based LAI." This is an incomplete statement. You repeated this statement problem in the introduction, page 2, row 11. Maybe you meant to say "using" instead of "assuming".

Response: We do assume the satellite-based LAI data for the OSSE, but more precisely, we "simulate" the satellite-based LAI in the OSSE. Therefore, we will replace "assuming" with "simulating" in the revised manuscript.

Page 2. row 8. "straightforward" may be replaced by "numerically straightforward". In this context, it is not simple to go from local to global because spatial covariances become relevant.

Response: We agree, and will revise it accordingly.

Page 2. (last) row 31. "phase space stays the same" may be "phase space dimension stays the same"

Response: We will revise it accordingly.

---

## Author Response (AR1)

**Response to Referee #1 (Assistant Professor, Matthias Morzfeld)**

General comments: The paper presents a description of the application of a particle filter to assimilate leaf are index observations into a global vegetation model. The paper is concise, and presents only the necessary information, no long background/review section is provided. I find the numerical experiments and results convincing (pending specific comments below).

Response: Thank you very much for the useful, constructive comments to improve the paper. We revised the manuscript accordingly. Our point-by-point responses are shown in blue.

My main concern is that this paper might not be suitable for Nonlinear Processes in Geophysics. The reason is that the journal emphasizes new methods, applied to realistic problems. The paper simply presents an "old" method applied to a new problem. I find it interesting to read that a particle filter can solve an important and "real" data assimilation, however the general NPG readership might get bored. The authors should decide wether NPG is the best journal to reach the audience they want to reach. This is also reflected by the references, only few of which are to articles in journals similar to NPG. I suspect that this paper would also make a fine contribution in a journal that is more focused on, e.g., Earth system modeling. The authors may want to consider going that route.

Response: We would really appreciate the kind suggestion. After carefully considering other potential options, we reached to the conclusion that we would like to have this paper published in NPG because the main focus of this paper is methodology. It would not be trivial if well-known, "old" methods work with a new problem. As we mentioned in the discussion paper (P6. Line 26-27), "To the best of the authors' knowledge, this is the first study to assimilate the fine time-scale satellite data with an *individual-based* DGVM". We also explored sensitivities to the filter parameters such as the initial and resampling perturbation sizes and particle size. In the revised manuscript, Section 4 was added about the sensitivity experiments. We believe these methodological explorations would be useful for the readers of NPG. In addition, this study performs simple experiments at a single location with only a couple of plant functional types (PFTs). Future studies will include spatial distributions with more PFTs, and will be more relevant to domain-specific journals. We added related descriptions in the revised manuscript (P.2 Line 8-11, P9.

Specific comments:

1. I wonder if there is any sensitivity to how repeated particles are perturbed after resampling. The authors chose a random perturbation, but miss to motivate their choice. I think the paper should contain numerical experiments where it is shown that either the method is robust to (small) changes in how repeated particles are perturbed, or it should be reported how the perturbations influence the results.

Response: Following the suggestion, we performed additional experiments with different random perturbation settings, and found that the filter collapsed for biomass with smaller perturbation settings, though estimated model parameters and other state variables were estimated accurately without collapse. We added a new section in the revised manuscript to show these results and to discuss the sensitivity to the perturbation settings (P7. Line 12 -21).

2. The number of particle used is typically important for the results one obtains with a particle filter. Indeed, much of the meteorological literature says that the number of particles required is excessive. To address this issue, I would suggest to run more numerical experiments with a varying number of particles. One can then compute, e.g., means and variances, and check that the method has converged when, e.g., 8000 particles are used. Specifically, I suggest experiments with 4000, 8000 and perhaps 16000 particles (if possible).

Response: Following the suggestion, we performed additional experiments with different particle sizes ranging from 500 to 16000. The results showed that the filter collapsed for biomass with 4000 particles or less when we keep the same random perturbation setting. Also, the model parameter and other state variable were not estimated accurately with 500 particles with a smaller random perturbation setting. We included the sensitivity to the particle sizes along with the sensitivity to the random perturbation settings (cf. previous comment) in the revised manuscript (P7. Line 12 -21).

3. I wonder what happens when the data assimilation is initialized with a "smaller" initial uncertainty. The authors define intervals for the parameters, but do not mention how they came up with these intervals. It would be interesting to see what happens when these intervals are shortened or widened. In particular, the particle filter has no mechanism to bring the parameters to values that are not contained within the initial set. This could make things difficult for the

"real life" application. Again, I suggest to investigate this issue with more numerical experiments.

Response: We would appreciate the suggestion to perform additional sensitivity experiments on the initial parameter uncertainties. We selected the intervals of the initial parameters based on the ecological knowledge from the previous studies (Kolari et al., 2006; Zeng et al., 2011; Zhao et al., 2015; Takagi et al., 2015). We added the specific references in the revised manuscript (P5. Line 8-9).

Following the comment, we performed additional experiments with different initial parameter uncertainties. The results showed that the parameters were not estimated accurately with wider initial intervals when 2000 particles or less are used, although they were estimated accurately with narrower initial intervals. With 8000 particles, the parameters were estimated accurately even with wider initial intervals. Sampling a wider interval with a smaller particle size generally reduces the particle density, or the effective number of particles, so that the results seem to be reasonable. We included these sensitivity results in the revised manuscript (P6. Line 29 - P7. Line 10).

4. In figs.4 (right column), 5d (right column), 7 (right column), and 8d (right column): it seems that the data assimilation only impacts the parameter estimates for parts of the year, however data are assimilated every 4 days. The authors miss to provide a clear explanation of why that is the case.

Response: We assimilated the LAI only when 0.5 or larger, therefore, data assimilation has impacts only in the summer when the leaves appear (i.e., LAI $\geq$ 0.5). We added this discussion in the revised manuscript (P5. Line 33 - P6. Line 1).

5. I would remove all NODA figures, as they do not really carry information. It is clear that when no data assimilation is used, no parameter is changed.

Response: We believe that it would be important to show NODA figures to highlight the impact of data assimilation. Since this is the first time to apply DA to SEIB-DGVM, it is unclear what impact DA would bring. Therefore, we would really like to keep the side-by-side comparison of the experiments with and without DA.

Technical corrections:

I find the use of "newly" in the first sentence of the abstract a bit unusual. I would suggest to re-formulate this sentence. The sentence also appears again later on (p.2. line 6, p.6 line 27), and there it should also be changed.

Response: We corrected these sentences in the revised manuscript (P1. Line 11, P2. Line 7).

Response to Referee #2 (Dr. Malaquias Pena Mendez)

General comments:
1. This paper applies a particle filter data assimilation scheme to assimilate MODIS Leaf Area Index (LAI) data into an explicit individual-plants dynamical global vegetation model. Results indicate that the scheme reduces the uncertainty of the LAI analyses as compared to random initialization. Furthermore, the technique appears to successfully estimate the model parameters that control separately LAI for the forest and for the grass types, out of whole LAI observations.
2. The content of the paper is relevant for Earth Systems and non-linear modeling. It addresses one important aspect to increase models realism through the use of information contained in fine time-scale observed data.
3. The problem addressed in this study is challenging considering the nonlinearities in the dynamics of the vegetation, the multitude of interactive physical and biogeochemical processes taking place at the local and regional scales, and the fact that not all state variables are observed or retrieved by satellite.
4. The study is well executed to proof the concept with all the needed elements (calibrated model, quality controlled data, an optimized data assimilation scheme) and reduced (only a few geographical points) scope to make it successful.

Response: Thank you very much for the useful, constructive comments to improve the paper. We revised the manuscript accordingly. Our point-by-point responses are shown in blue.

Specific comments and questions:
1. While the description is succinct and easy to follow, it needs to make explicit major assumptions made and the problems one may encounter if they were to be relaxed.

Response: Thank you very much for the suggestion. We made strong assumptions in the OSSE, and relaxed some in the real-world experiment. We still made strong assumptions in the limited area application. We made these assumptions clearer in the revised manuscript (P9. Line 11-13).

2. Below is a list of questions that arose while reading the manuscript:
2.1: What modifications to the original vegetation model were made to adapt it to the DA scheme? Were these only the changes in parameter values we see in the appendix?

Response: The model simulates daily states, but the original model outputs were only once per year. Daily outputs are needed for data assimilation every once in four days. Therefore, we modified the model code to output the daily states every 4 days. In addition, the original model code assumed running for many years continuously, and the initial seed for the random number generator was fixed. Since in this study we stopped the model every 4 days, and the same seed was repeated every time we started the model. Therefore, we modified the model code to randomly generate the seed for the random number generator every time when we initiate the model. These are the only modifications not shown in the paper, because we thought these were only minor technical modifications. To explicitly describe all necessary changes to the existing model code, we included these in the revised manuscript (P2. Line 25-31).

2.2: Are the field observations in the Siberia Yakutsk Larch forest site independent or were they included to create the climate forcing data 2001-2007 in the vegetation model?

Response: The climate forcing data were created using the NCEP/NCAR reanalysis data and CRU observation based data. The observed climate data at this site were not directly used in our experiments, but these data might be included in the NCEP/NCAR reanalysis data. It is not simple to find if the site observation data were reported through GTS and included in the NCEP/NCAR reanalysis. Therefore, we included a sentence in the revised manuscript describing the possibility that the observations at the site may be used in the NCEP/NCAR reanalysis through GTS (P4. Line 22-23).

2.3: Was the 2004-2007 period of MODIS 4-days frequency data continuous on the study site? Were there missing data? How was the missing data handled?

Response: There are a number of missing data in the quality-controlled MODIS data. Therefore, as we have described in P5. Line 30-31 in the discussion paper, if the number of the quality controlled MODIS data in the 10-km radius contains less than 300 grid points, we set these data as the missing data. We revised the manuscript to describe more explicitly about the missing data from the original quality-controlled MODIS data (P7. Line 31 - P8. Line 2).

2.4: Was the 8000 particles generated decided by computer capacity, or any other criteria?

Response: In response to the other reviewer's comment #2, we performed additional

experiments with different particle sizes. We added a new section to show the sensitivity to the particle size in the revised manuscript (P6. Line 29 - P7. Line 21).

2.5: Simulated observations (in the OSSE) versus real observations: How do they compare? Were the real observations also normally distributed? Were standard deviations of real observations about 10% as in the OSSE experiment?

Response: We assumed the normal distribution for the real observation error. The error standard deviation is included in the MODIS dataset that we used (Knyazikhin et al., 1999). As already described in P.6 Lines 1-2 in the discussion paper, we used "the median of the error standard deviations" in the 10km radius. The standard deviations of the real observations are different from those of the OSSE, as indicated by Figs. 3-a and 6-a. We explicitly described about the differences of the observation error standard deviations between the simulated and real observations in the revised manuscript (P8. Line 3-6).

2.6: Did you follow any particular rule to determine the perturbation size of Pmax and Dor? In the study you allow larger amplitude perturbations for forest than for grass types. The amplitude of Dor is relatively very small.

Response: There are two perturbation settings for the model parameters: the initial perturbation sizes and the random perturbation sizes when resampling. We selected the initial perturbation sizes based on the ecological knowledge from the previous studies (Kolari et al., 2006; Zeng et al., 2011; Zhao et al., 2015; Takagi et al., 2015). The initial Pmax perturbation size for grass is 4 times smaller than that of forest. The initial Dor perturbation sizes for grass and forest are the same. The random perturbation sizes when resampling follow the initial perturbation sizes. We added the references and explicit descriptions about the perturbation settings in the revised manuscript (P5. Line 8-9, P5. Line18-19).

Following the comment of the other referee, we performed additional experiments with different random perturbation settings for the initial perturbation sizes and the random perturbation sizes when resampling. We added a new section to show these results and to discuss the sensitivity to the perturbation settings in the revised manuscript (P6. Line29 - P7. Line 21).

2.7: The manuscript indicates that perturbations of parameters are applied only to duplicated particles. Since the particle DA scheme eliminates particles far away from observations (Fig. 1),

that would mean that the range of the distribution of all the particles decreases after several cycles at least compared to the initial (uniform) distribution. Is this correct? Still, you do not report any issue with collapsing of the DA scheme when observations are outside the range of the distribution of particles. Can you please, elaborate more on this issue?

Response: Yes, it is correct that the range of the distribution of all the particles decreases after several cycles. If we apply proper random perturbations to the duplicated particles, we can avoid filter collapse. However, our additional experiments showed filter collapse when the particle size is 500. We described about the collapse in the new section on the sensitivity to the particle size and random perturbation size (P6. Line29 - P7. Line 21).

2.8: The NODA and the TEST experiments; Figure 3. How the 8000 particles are inserted at the initial conditions? Is this done every 4 days with a uniform distribution each time? The TEST experiment appears to reduce a big systematic error that appear during the growing months. Traditional DA schemes apply a bias-correction strategy of the First Guess prior to performing the analysis. Does this mean that particle DA also removes systematic errors?

Response: As already described in P.4 Line 28 in the discussion paper, "The 8000 particles at the end of the 103-year spin-up runs are used as the initial conditions for DA", and the NODA and TEST experiments start from the same initial 8000 particles. The 8000 particles continue to be the same until the first observation of LAI is assimilated. Since the LAI is observed only when 0.5 or larger, the LAI observation exists only in the summer season. Model state and parameters are estimated together at DA, and the model systematic errors associated with the parameters are corrected by DA with parameter estimation. No explicit bias correction is applied. We added these descriptions (P5. Line 4-5, P5. Line 14-17)

The systematic errors in NODA comes from the uncertain parameter settings. TEST can estimate parameters using observed LAI, and therefore, can reduce the systematic errors. This is different from the bias-correction strategy of the first guess. There is no explicit bias correction applied to the TEST experiment. So, we understand that this particular particle DA can reduce systematic errors by estimating the uncertain model parameters. We discussed this point in the revised manuscript (P6. Line 6-8).

2.9: TEST experiment; Figure 3a (forecast+grass). Please, explain the problem at the end of the fall months (circled in blue in the attached figure). Can this be attributed to neglecting observations when LAI<0.5 ? Will this be removed if observations are added there?

Response: Yes, because we assume observation to be available only when LAI≥0.5, it is difficult to estimate the LAI when observed LAI is less than 0.5. We did not perform experiments with small LAI observations, because the MODIS data for the real-world experiment did not include LAI < 0.5 (Fig. 6). There are too few data with real MODIS LAI < 0.5, and our preprocessing assigns the missing value. We added these discussions in the revised manuscript (P5. Line 3-5).

2.10: It is obscure to me how come the individual LAI of Forest and Grass are accurately estimated out of the whole LAI. Even when the whole LAI estimation is incorrect as in the periods in the blue circle in the attached figure. What mechanism or statistical assumption within the DA process makes the partitioning of LAI correct? Is this pure chance?

Response: Thank you for the comment, which initiated further analysis of the results that were not shown in the manuscript. As already described in P.5, Lines 21-22 in the discussion paper, "To investigate the sensitivity to the choice of the nature run, we performed similar OSSEs by replacing the nature run with other randomly-chosen parameter sets". We investigated these different OSSEs more carefully and found that the parameters for grass were estimated well when the nature run used a larger Pmax value for grass. However, in another OSSE, the nature run used a small Pmax value; the results showed that the parameters for grass showed significantly larger uncertainties, while the parameters for forest were estimated well. Larger Pmax values for grass produce more grass LAI, which can be observed with the observing threshold of LAI = 0.5 near the growing and falling periods (shown by the blue circles provided by the reviewer). With smaller Pmax values for grass, the small grass LAI cannot be observed directly, but the large LAI observations in the summer season predominantly suggest forest LAI. This would allow to estimate the forest parameters well, although the grass parameters showed larger uncertainties. We included these results and discussions in the revised manuscript (P6. Line 14-27).

2.10: In the Real-World Experiment; there is no detail on the perturbation strategy, so I suppose it is the same as in the OSSE experiment.

Response: We did not provide the details of the experimental settings. Yes, the perturbation strategy of the real-world experiment is same as that of the OSSE. We added the descriptions in the revised manuscript (P7. Line 24-27).

2.11: The observation error standard deviation in the real case needs more explanation. What is the truth from which the error is estimated? Is this the in situ observation? Is this error an input in the DA scheme?

Response: As described in our previous response to the comment #2.5, P6 lines 1-2 in the discussion paper reads "The observation error standard deviations are assigned to each LAI datum from the original source, and we took the median of the error standard deviations." We rely on the original MODIS data source about the estimate of the observation error standard deviation. We revised this sentence about the observation error standard deviations to avoid potential misunderstanding (P8. Line 3-6).

Technical corrections
Page 1. Abstract. ".. newly developed" should be ".. developed". You repeated that later on in the text.

Response: We corrected it accordingly (P1. Line 11, P2. Line 7).

Page 1. Abstract. ".., assuming the satellite-based LAI." This is an incomplete statement. You repeated this statement problem in the introduction, page 2, row 11. Maybe you meant to say "using" instead of "assuming".

Response: We do assume the satellite-based LAI data for the OSSE, but more precisely, we "simulate" the satellite-based LAI in the OSSE. Therefore, we replaced "assuming" with "simulating" in the revised manuscript (P1. Line 13, P2. Line 14, P5. Line 2).

Page 2. row 8. "straightforward" may be replaced by "numerically straightforward". In this context, it is not simple to go from local to global because spatial covariances become relevant.

Response: We agree, and revised it accordingly (P2. Line11).

Page 2. (last) row 31. "phase space stays the same" may be "phase space dimension stays the same"

Response: We revised it accordingly (P3. Lines 10, 17).

[revised manuscript text omitted]

---

## Author Response (AR2)

Response to Editor (Associate professor Amit Apte)

Comments to the Author:
Dear Authors, Thanks a lot for the detailed replies to the first set of reviews and also the major revisions. As the Report #1 below shows, there are some points that still need further modifications to the manuscript. I would request you to revise the manuscript to take care of these comments and submit a revised version. Thanks.

Non-public comments to the Author:
I would like to request you to address the reviewer comments in report #1, after which I will send the manuscript to only one reviewer (the first one). I would also like to request you to add a paragraph about the size of the model state space (or the range of sizes, since each particle has different state space) and about the type of nonlinearities present in the model. That will help the reader appreciate the significance of results. Thanks.

Response: Thank you very much for the constructive comments to improve the paper. We revised the manuscript according to the comments by Referee #1. We also added a paragraph about the size of the model state space (P2. Line 31- P3. Line 5). According to the definitions of "forest" and "tree" in this paragraph, we used these terms properly throughout the revised manuscript. Nonlinearities present in the model is described in P3. Line 12-13.

**Response to Referee #1 (Assistant Professor, Matthias Morzfeld)**

Suggestions for revision or reasons for rejection:

I thank the authors for addressing my previous concern about the manuscript. The additional numerical experiments are indeed very useful in accessing the validity of the approach and of the conclusions of this paper.

Response: Thank you very much for the useful, constructive comments to improve the paper. We revised the manuscript accordingly. Our point-by-point responses are shown in blue.

The authors state that they use an "efficient particle filter approach", sequential importance resampling (SIR). However, it is widely known and generally accepted that particle filters are not efficient in the sense that an (usually) excessive amount of particles is needed to solve a given problem, especially if the dimension of the system is large. If the number of particles is large, the computations required to perform particle filtering also become large, as each particle requires (at least) one simulation.

Response: Thank you very much for the suggestion. We tested SIR as the first attempt to construct the DA system with SEIB-DGVM, however it is not an "efficient" approach when the dimension of the system is large. Following the suggestion, we modified the description of the particle filter (P3. Line 23-25).

This failure of particle filters can be seen in this application, even though relatively large numbers of particles (500-16,000) are used. The failure becomes apparent when looking at figures 4, 6, 8, where the mean and "error bars" are shown for the estimated parameters. It can be seen that the data assimilation by particle filters changes the mean and decreases the variance, however the variance decrease is largely dominated by the amount by which resampled particles are perturbed. These perturbations are required to avoid "particle filter collapse", as is shown by the numerical experiments with smaller (and larger) perturbations. However, the collapse is only avoided by allowing the particles to "spread out" to within the bounds dictated by the perturbations. These bounds are somewhat arbitrary, and, for that reason, the results and conclusions are also somewhat arbitrary. Specifically, I expect that figures 4, 6, 8 look very different when different perturbations are used during resampling. In short, the numerical experiments suggest that the particle filter picks "the best particle", i.e., the one that is closest to the

observation, and then defines uncertainty estimates that are largely dominated by the perturbations of the resampled ensemble.

The authors state in their conclusions that uncertainties of the state variables and model parameters are greatly reduced, and that, overall, uncertainty is significantly reduced. This is true, however the reduction is dominated by the perturbations applied to the resampled ensemble and the paper makes no suggestions as to how to chose these perturbations. It merely tests a few settings (small, moderate and large), and from these tests concludes that "moderate" is a good choice. In this way, the method yields results that are largely influenced by certain choices of parameters in the data assimilation method, rather than that the data assimilation method produces useful results without excessive tuning. On the other hand, many data assimilation in daily use, e.g., the ensemble Kalman filter are known to "work" only after extensive tuning. However, the tuning is motivated by the physics of the problem or corroborated by theory and extensive numerical experiments. I recommend that the authors be more cautious about the tuning choices they make, to explain better why these choices are necessary and how guidelines for this tuning process can be established, at least for this or closely related problems (such as the one that perform this data assimilation on a global scale).

Response: We would really appreciate the suggestion. As pointed out, we did not describe clearly how to choose the resampling perturbations, or we did not describe precisely the limitation of this study. Therefore, we added the detailed explanations to the revised manuscript as follows.

In this study, biomass was the most sensitive to the resampling perturbation sizes. When resampling, the random perturbations were applied only to Pmax and Dor, not to the model state variables. This contributes to reduce the variety of vegetation structures such as tree densities and tree heights due to the frequent DA every four days (not shown). This tends to cause filter divergence for biomass even for a large particle sizes when the resampling perturbation size is small (Table5). When the perturbation size is relatively large, degeneracy of the vegetation structure is mitigated to some extent. After some tuning, we found proper perturbation sizes that work for stable filtering without causing particle degeneracy, especially for biomass which is found to be the most sensitive to the perturbation sizes. We added more detailed descriptions about the necessity to tune the resampling perturbation sizes especially for biomass (P5. Line 24-26, 28-29, P7. Line24-25, P9. Line 20-27).

To avoid the filter collapse for biomass, we used the "moderate" perturbation size for the TEST and the real-world experiment. However, the moderate perturbation size may be large for variables other than biomass, and this may be why the estimated parameters show occasional jumps. Table 5 also shows that, with 4000 particles or more, the parameters and state variables except for biomass were estimated accurately, although the filter collapsed for biomass with smaller perturbations even for large particle sizes. We added the description of the difference of the sensitivities (P7. Line 32 - P8. Line 2).

If the resampling perturbation size is small enough, the artificial spread caused by the resampling perturbations can be ignored. However, the relatively large perturbations used in this study dominated the uncertainty estimates, as suggested by the reviewer. We agree that the concluding description was not accurate and may be misleading. Therefore, we removed the misleading expression (P. 9 Line 12-13).

As the first attempt to construct the DA system with SEIB-DGVM, we simply added perturbations to Pmax and Dor, however adding resampling perturbations to other variables would be better. In the future study, we will explore more effective resampling methods to avoid the filter collapse for biomass. We added the limitation of this study and necessity of the modification of the DA system in the future study (P9. Line 27-30).

[revised manuscript text omitted]

---

## Author Response (AR3)

**Response to Editor (Associate professor Amit Apte)**

Dear authors, Since you have addressed all the comments from both the referees, I am happy to recommend the paper for publication subject to a few small modifications suggested by the referee, including the citations to works about failure of particle filter in high dimensional problems. Since these are quite minor changes, I am recommending "publication subject to technical corrections" without any further review, so it is up to you to decide whether these modifications are needed or not. Thanks.

Response: Thank you very much. We revised the manuscript according to the comments by Referee #1.

**Response to Referee #1 (Assistant Professor, Matthias Morzfeld)**

I thank the authors for addressing my concerns and apologize for the delay in my review. I suggest the authors make two minor modifications to the manuscript.

Response: Thank you very much for the useful, constructive comments to improve the paper. We revised the manuscript accordingly. Our point-by-point responses are shown in blue.

(1) Since the perturbations after resampling may influence the covariance estimates after each assimilation step, I suggest to add a few words of caution that this DA technique (as is) does not necessarily provide accurate estimates of the errors one can expect after data assimilation.

Response: We added descriptions about the limitation to error estimate (P9. Line 32 – P10. Line 3).

(2) The particle filter is used to estimate four parameters. In that sense, the "dimension" of the DA problem is not very large (even though the state dimension is in the thousands). This small dimension may be another important factor in why the particle filter works well in this application. The authors can back up their claim that particle filter do not work so well in high-dimensional problems using the following references:

(i) C. Snyder, T. Bengtsson and M. Morzfeld, Performance bounds for particle filters using the optimal proposal, Monthly Weather Review 143, 4750 – 4761 (2015).
(ii) Bickel, P., B. Li, and T. Bengtsson, 2008: Sharp failure rates for the bootstrap particle filter in high dimensions. Pushing the Limits of Contemporarly Statistics: Contributions in Honor of Jayanta K. Ghosh, B. Clarke and S. Ghosal, Eds., Vol. 3, Institute of Mathematical Statistics, 318–329
(iii) Snyder, C., 2012: Particle filters, the "optimal" proposal and high-dimensional systems. ECMWF Seminar on Data Assimilation for Atmosphere and Ocean, Shinfield, United Kingdom, ECMWF, 161–170.
(iv) Snyder, C., T. Bengtsson, P. Bickel, and J. Anderson, 2008: Obstacles to high-dimensional particle filtering. Mon. Wea. Rev., 136, 4629–4640.

Response: We added the references and related discussions accordingly (P3. Line 24, P9.

Line 17-22).

[revised manuscript text omitted]